# DeepSearch: Overcome the Bottleneck of Reinforcement Learning with Verifiable Rewards via Tree-based Search

**Fang Wu**$^{\heartsuit *}$   **Weihao Xuan**$^{\triangledown, \triangle *}$   **Heli Qi**$^{\triangle *}$   **Aaron Tu**$^{\spadesuit}$   **Ximing Lu**$^{\diamondsuit}$
**Li Erran Li**$^{\clubsuit}$   **Yejin Choi**$^{\heartsuit \dagger}$
$^{\heartsuit}$Stanford University $^{\triangledown}$University of Tokyo $^{\triangle}$RIKEN AIP $^{\spadesuit}$UC Berkeley
$^{\diamondsuit}$University of Washington $^{\clubsuit}$Amazon AWS

## Abstract

Although Reinforcement Learning with Verifiable Rewards (RLVR) has become an essential component for developing advanced reasoning skills in language models, contemporary studies have documented training plateaus after thousands of optimization steps, i.e., notable decreases in performance gains despite increased computational investment. This limitation stems from the sparse exploration patterns inherent in current RLVR practices, where models rely on limited rollouts that often miss critical reasoning paths and fail to provide systematic coverage of the solution space. We present DeepSearch, a framework that integrates Monte Carlo Tree Search (MCTS) directly into RLVR training. In contrast to existing methods that rely on tree search only at inference, DeepSearch embeds structured search into the training loop, enabling systematic exploration and fine-grained credit assignment across reasoning steps. Through training-time exploration, DeepSearch addresses the fundamental bottleneck of insufficient exploration, which leads to diminishing performance gains over prolonged training. Our contributions include: (1) a global frontier selection strategy that prioritizes promising nodes across the search tree, (2) selection with entropy-based guidance that identifies confident paths for supervision, and (3) adaptive replay buffer training with solution caching for efficiency. Experiments on mathematical reasoning benchmarks show that DeepSearch achieves an average accuracy of 62.95% and establishes a new state-of-the-art reasoning model, while using 5.7x fewer GPU hours than extended training approaches. These results highlight the importance of strategic exploration over brute-force scaling and demonstrate the promise of algorithmic innovation for advancing RLVR methodologies. DeepSearch establishes a new direction for scaling reasoning capabilities through systematic search rather than prolonged computation.

 https://github.com/smiles724/DeepSearch

 https://huggingface.co/fangwu97/DeepSearch-1.5B

## 1 Introduction

Large language models (LLMs) have recently achieved notable progress on complex reasoning tasks (DeepSeek-AI, 2025; Yang et al., 2024; Wu et al., 2025a; Wang et al., 2026; Xia et al., 2025), driven in part by test-time computation scaling strategies (Li et al., 2023; Yao et al., 2023; Bi et al., 2024; Zhang et al., 2024a; Guan et al., 2025) such as tree search with process-level evaluation. While effective, these methods typically treat structured search as an inference-only mechanism, leaving untapped potential to integrate systematic exploration into the training process itself.

This separation between training and inference imposes fundamental limitations on the scalability of reinforcement learning with verifiable rewards (RLVR) for reasoning. Current RLVR approaches

---

$^{*}$Equal contributions.
$^{\dagger}$Corresponding author. Email: yejinc@stanford.edu

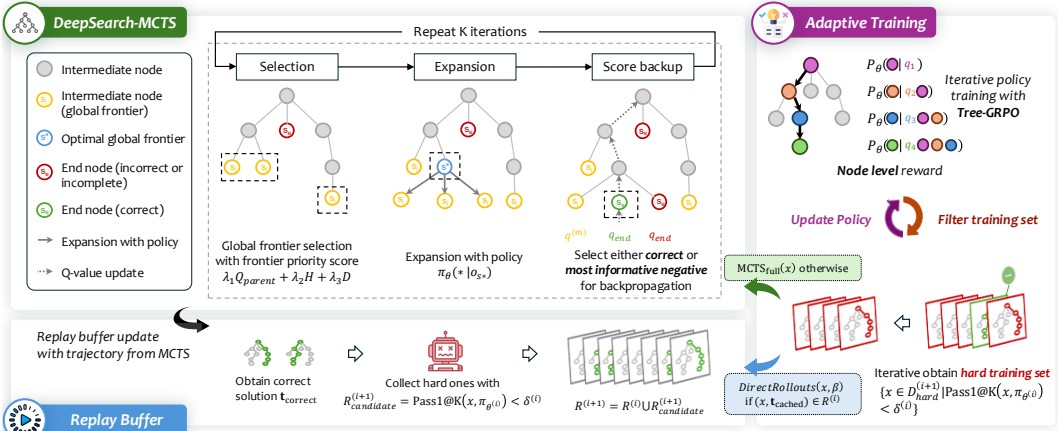

Figure 1: DeepSearch framework overview with three key components.

remain constrained by sparse exploration patterns during training (Wu et al., 2025b; Liu et al., 2025c; Tu et al., 2025), while models are expected to demonstrate sophisticated search behaviors only at inference time. Even recent advances in prolonged RL training (Liu et al., 2025a) have shown that performance plateaus after thousands of steps, with diminishing returns to allocating more compute to deeper training. This suggests that simply scaling the number of training steps, the primary axis explored in prior work, may not be sufficient to fully realize RLVR's potential.

We address this gap by introducing DeepSearch, a framework that embeds Monte Carlo Tree Search (MCTS) (Metropolis & Ulam, 1949) directly into RLVR training, representing a fundamental shift from scaling training depth to scaling training breadth. By coupling structured search with verifiable rewards during training, DeepSearch enables models to learn not only from correct solutions but also from the systematic exploration process itself, providing richer supervision than outcome-based or direct rollout methods (Lyu et al., 2025; He et al., 2025b).

The core insight driving us is to *focus on training-time exploration* as the driver of improved reasoning. While traditional RLVR relies on limited rollouts that may miss critical reasoning paths, DeepSearch systematically expands the reasoning frontier during training through principled tree search. This design advances three key objectives: *(i)* expanding reasoning coverage beyond what direct policy rollouts can achieve, *(ii)* providing fine-grained credit assignment to intermediate reasoning steps through tree-structured backpropagation, and *(iii)* maintaining computational efficiency through intelligent node selection and solution caching strategies.

Towards these goals, DeepSearch introduces several key innovations. First, *global frontier selection* strategy prioritizes the most promising nodes across the entire search tree, moving beyond traditional root-to-leaf Upper Confidence Bounds for Trees (UCT) traversals that can be computationally wasteful and myopic. Second, *selection with entropy-based guidance* systematically identifies confident incorrect reasoning paths for supervision. Finally, an adaptive training strategy with replay buffers progressively filters challenging problems and caches verified solutions, thereby avoiding redundant computation across training iterations.

We evaluate DeepSearch on math reasoning benchmarks, where it significantly outperforms strong RLVR baselines (Liu et al., 2025a; Luo et al., 2025b). Our results show that DeepSearch achieves 62.95% average accuracy on challenging mathematical tasks, representing **a new state-of-the-art for 1.5B reasoning models**. Importantly, these gains are achieved while maintaining computational efficiency through progressive filtering and intelligent reuse of solutions, demonstrating that search-augmented training can be both more effective and more practical than conventional approaches.

The implications extend beyond math reasoning: by bridging the gap between inference-time search capabilities and training-time learning, DeepSearch establishes a new approach for scaling RLVR that emphasizes systematic exploration over prolonged training. This work suggests that the future of reasoning model development lies not just in scaling model parameters or training steps, but in fundamentally rethinking how we structure the learning process to mirror the sophisticated reasoning patterns we expect at inference time.

## 2 RELATED WORKS

**Search-based reasoning.** Structured search has become a standard strategy for scaling test-time compute in LLMs (Snell et al., 2024; Wu et al., 2024; Zhang et al., 2024c), with diverse methods including tree-based (Yao et al., 2023; Zhang et al., 2024b; Qi et al., 2024) and random sampling approaches (Wang et al., 2022). Recently, search-based reasoning has evolved into sophisticated frameworks that integrate three core components: policy models, reward models, and search algorithms. Drawing inspiration from game-playing systems like AlphaGo (Silver et al., 2016), works have explored Monte Carlo Tree Search (MCTS) and beam search to guide LLMs through structured reasoning processes (Chen et al., 2024; Zhang et al., 2024a;c), particularly following OpenAI's o1 release (Jaech et al., 2024). These frameworks enable exploration of multiple solution paths during inference, trading compute resources for improved accuracy on challenging tasks such as math reasoning. Key design considerations include outcome-supervised versus process-supervised reward models, discriminative versus generative reward architectures, and search strategies ranging from local selection to global exploration (Lightman et al., 2023; Wang et al., 2023). However, most methods restrict search to inference and do not integrate exploration signals into training, leaving the potential for jointly optimizing search and learning largely unexplored.

**Reinforcement learning from verifiable rewards.** RLVR has emerged as a transformative approach for aligning and enhancing LLMs by addressing critical challenges across instruction following (Su et al., 2025; Gunjal et al., 2025), ethical alignment (Wang et al., 2025a), and reasoning capabilities (Wang et al., 2025b). Recent extensions (Guo et al., 2025; Yu et al., 2025; Wan et al., 2025) have improved training stability and efficiency by incorporating critic-free optimization, dynamic sampling, and adaptive weighting mechanisms. While these approaches demonstrate the promise of RLVR, they predominantly rely on direct rollouts, which can constrain systematic exploration of the solution space (Wu et al., 2025b; Yue et al., 2025).

**Monte Carlo Tree Search.** MCTS is a powerful search paradigm for complex decision-making problems, extensively explored across diverse fields like games (Silver et al., 2016; Ye et al., 2021), robotics (Best et al., 2019; Dam et al., 2022), theorem proving (Lample et al., 2022), and matrix multiplication (Fawzi et al., 2022). Early work, such as AlphaGo, integrated MCTS with deep learning (Kemmerling et al., 2023), achieving superhuman performance in board and video games (Ye et al., 2021). Recently, MCTS has been applied to path finding and train timetabling problems (Pitanov et al., 2023; Yang, 2023), while Vagadia et al. (2024) integrated MCTS into physics-informed planning networks for robot control. Despite the demonstrated potential of MCTS for heuristic exploration, it remains unclear how to effectively employ it during RLVR training.

## 3 DEEPSEARCH WITH MCTS

Given a problem $x$ and a policy model $\pi_\theta$, we adopt a modified MCTS framework to build a search tree for incremental step-by-step solution exploration. We replace traditional root-to-leaf selection with global frontier-based node selection. The root node represents the question $x$, and child nodes correspond to intermediate steps $s$ generated by $\pi_\theta$. A root-to-leaf path ending at a terminal node $s_{\text{end}}$ forms a trajectory $\mathbf{t} = x \oplus s_1 \oplus s_2 \oplus \ldots \oplus s_{\text{end}}$, where each step $s_i$ is assigned a q-value $q(s_i)$. Then we extract solution trajectories $\mathbb{T} = \left\{ \mathbf{t}^1, \mathbf{t}^2, \ldots, \mathbf{t}^n \right\}$ $(n \geq 1)$ from the search tree $\mathcal{T}$, where $\mathbf{t}^i$ can be correct, incorrect or incomplete. The depth of any node $s$ is denoted as $d(s) \in \mathbb{Z}^+$. $N(s)$ and $\xi(s)$ denote the number of visits to $s$ and the number of children nodes of $s$, respectively. Starting from the root node $x$, our MCTS iterations are conducted through three subsequent components.

### 3.1 EXPANSION WITH ENTROPY-BASED GUIDANCE

In step $i$, we collect the latest reasoning trajectory $o_i = x \oplus s_1 \oplus s_2 \oplus \ldots \oplus s_{i-1}$ as the current state, i.e., observation. Based on this state, we prompt the policy model $\pi_\theta(s_i|o_i)$ to generate $n$ candidates for the next-step reasoning trail $\{s_{i,j}\}_{j=1}^n$. We repeat this expansion behavior until we reach the terminal nodes $s_{\text{end}} \in \mathcal{S}_{\text{end}}$, either by arriving at the final answers or by hitting the maximum depth of the tree $d_{\mathcal{T}}$, which yields an ordered sequence $s_1 \rightarrow \cdots \rightarrow s_{\text{end}}$.

During each expansion, let $\mathcal{S}_{\text{end}}^{(k)}$ denote the set of newly generated terminal nodes at iteration $k$. We evaluate the correctness of each terminal node using a verification function $\mathcal{V} : \mathcal{S}_{\text{end}} \rightarrow \{0, 1\}$, where $\mathcal{V}(s) = 1$ indicates a correct solution and $\mathcal{V}(s) = 0$ indicates an incorrect or incomplete

solution. Then we partition the terminal nodes into correct and incorrect/incomplete subsets:

$$\mathcal{S}_{\text{correct}}^{(k)} = \{s \in \mathcal{S}_{\text{end}}^{(k)} \mid \mathcal{V}(s) = 1\}, \quad \mathcal{S}_{\text{incorrect}}^{(k)} = \{s \in \mathcal{S}_{\text{end}}^{(k)} \mid \mathcal{V}(s) = 0\}. \tag{1}$$

If $\mathcal{S}_{\text{correct}}^{(k)} = \emptyset$, we employ an *entropy-based selection* to identify the most confident wrong rollout. The terminal node with the lowest average entropy along its root-to-leaf trajectory is selected:

$$s_{\text{neg}}^* = \arg \min_{s \in \mathcal{S}_{\text{incorrect}}^{(k)}} \bar{H}(\mathbf{t}(s)), \tag{2}$$

where $\mathbf{t}(s) = (x, s_1, s_2, \ldots, s)$ represents the unique trajectory from root $x$ to terminal node $s$, and the average trajectory entropy is defined as $\bar{H}(\mathbf{t}(s)) = \frac{1}{|\mathbf{t}(s)|} \sum_{i=1}^{|\mathbf{t}(s)|} H(\pi_\theta(s_i \mid o_i))$,, where $H(\pi_\theta(s_i \mid o_i)) = -\sum_{a_{i,k}} \pi_\theta(a_{i,k} \mid o_i, a_{i,<k}) \log \pi_\theta(a_{i,k} \mid o_i, a_{i,<k})$ is the Monte Carlo estimation of the Shannon entropy of the token distribution at step $i$. $a_{i,k}$ is the $k$-th token of step $s_i$, and $a_{i,<k}$ denotes the tokens preceding $a_{i,k}$. This strategy prioritizes incorrect reasoning sequences with low decision uncertainty, targeting areas where the model is most confident in its decisions and would benefit from additional training and supervision. We find that this most-confident incorrect selection consistently outperforms random and least-confident selection across all benchmarks, see Table 5.

## 3.2 HEURISTIC SCORE BACKUP

Let $\mathbf{t}^*$ denote the selected trajectory for backpropagation, which is either a correct solution trajectory or the most confident negative trajectory $\mathbf{t}(s_{\text{neg}}^*)$ identified through entropy-based selection. Let $q^{(m)}(s_i)$ denote the *q-value* for node $s_i \in \mathbf{t}^*$ after the $m$-th rollout backpropagation. We define the iterative q-value update rule for nodes along the selected trajectory:

$$q^{(m)}(s_i) = q^{(m-1)}(s_i) + \gamma(i, l) \cdot q^{(m)}(s_{\text{end}}), \tag{3}$$

where $\gamma(i, l) : \mathbb{Z}^+ \times \mathbb{Z}^+ \to [0, 1]$ is the *depth decay* function that assigns higher weights to nodes closer to the terminal node. It is defined as $\gamma(i, l) = \max\left(\frac{i}{l}, \gamma_{\min}\right)$, where $i$ is the current node index in the trajectory, $l$ is the terminal node index, and $\gamma_{\min} = 0.1$ is the minimum decay threshold.

The q-value initialization is $q^{(0)}(s_i) = 0$ for all $s_i \in \mathcal{T}$. Terminal node rewards are assigned according to the verification function's result:

$$q(s_{\text{end}}) = \begin{cases} +1 & \text{if } \mathcal{V}(s_{\text{end}}) = 1 \text{ (correct)}, \\ -1 & \text{if } \mathcal{V}(s_{\text{end}}) = 0 \text{ (incorrect)} \vee d(s_{\text{end}}) < d_{\mathcal{T}} \text{ (incomplete)}. \end{cases} \tag{4}$$

To ensure positive q-values (*e.g.*, $q_{\text{correct}} = 0.1$) for nodes on correct reasoning paths while penalizing nodes leading to incorrect or incomplete solutions, we enforce the constrained update rule:

$$q^{(m)}(s_i) = \begin{cases} q^{(m-1)}(s_i) + \gamma(i, l) \cdot q^{(m)}(s_{\text{end}}) & \text{if } q^{(m-1)}(s_i) \cdot q^{(m)}(s_{\text{end}}) \geq 0, \\ \gamma(i, l) \cdot q^{(m)}(s_{\text{end}}) & \text{elif } q^{(m)}(s_{\text{end}}) > 0, \\ q^{(m-1)}(s_i) & \text{elif } q^{(m-1)}(s_i) > 0. \end{cases} \tag{5}$$

This constraint preserves the invariant that $q^{(m)}(s_i) \geq 0$ for all intermediate nodes $s_i \in \mathcal{T} \setminus \mathcal{S}_{\text{end}}$ leading to correct solutions, while allowing negative values only for nodes observed on any correct trajectory under the current search process. More justification is elucidated in Appendix B.2.

## 3.3 HYBRID SELECTION STRATEGY

UCT (Kocsis & Szepesvári, 2006) is the standard selection rule used in classical MCTS to balance exploitation of high-value nodes and exploration of under-visited ones. Our MCTS employs a *hybrid selection strategy* that combines traditional UCT-based local selection with novel global frontier selection, each serving distinct purposes in the search process.

**Local Selection for Sibling Comparison** During the expansion of a selected node, we generate multiple candidate children and must determine which to add to the tree. For this *local sibling comparison*, we follow the traditional MCTS protocol and employ the UCT algorithm as $\text{UCT}(s) = Q(s) + \lambda \sqrt{\frac{\ln N_{\text{parent}}(s)}{N(s)}}$, where $Q(s) = \frac{q(s)}{N(s)}$ represents the average reward per visit, $N_{\text{parent}}(s)$ is the number of visits from the parent node, and $\lambda$ balances exploitation and exploration. This local selection ensures that we make optimal decisions when choosing among sibling nodes that share the same parent and context.

**Global Frontier Selection for Next Expansion** After completing the first score backup phase, we need to identify the most promising node across the *entire search tree* for the next expansion round. This is where our novel global frontier selection mechanism operates.

Unlike traditional MCTS, which performs root-to-leaf traversals using UCT at each level, our global approach directly compares all frontier nodes simultaneously. We maintain a global view of all leaf nodes across the entire search tree $\mathcal{T}$ and prioritize promising expansion points globally:

$$\mathcal{F} = \{s \in \mathcal{T} \mid \xi(s) = 0, s \notin \mathcal{S}_{\text{end}}, d(s) < d_{\mathcal{T}}\}. \tag{6}$$

For each frontier node $s \in \mathcal{F}$ and its associated observation (prefix) $o_s$, we compute a *frontier priority score*:

$$F(s) = \underbrace{\lambda_1 \times \tanh(Q_{\text{parent}}(s))}_{\text{Quality Potential}} + \underbrace{\lambda_2 \times H(\pi_\theta(s \mid o_s))}_{\text{Uncertainty Bonus}} + \underbrace{\lambda_3 \times D(d(s))}_{\text{Depth Bonus}}. \tag{7}$$

Here, the quality potential term $\tanh(Q_{\text{parent}}(s))$ encourages the selection of nodes whose parents have demonstrated high value, using the tanh transformation to smoothly handle negative Q-values and map them to the range $[-1, 1]$. The uncertainty bonus term $H(\pi_\theta(s \mid o_s))$ provides exploration guidance by adjusting priority according to the policy's entropy; the sign of its coefficient can be utilized to steer selection toward regions with high confidence or uncertainty. The depth bonus term $D(d(s))$ encourages deeper exploration by providing additional priority to nodes at greater depths, where we empirically find $D(d(s)) = \sqrt{d(s)/d_{\mathcal{T}}}$ to be most effective among other variants including $d(s)$ and $\log(d(s) + 1)$. The node with the highest frontier score is selected for the next expansion: $s^* = \arg\max_{s \in \mathcal{F}} F(s)$.

**Rationale for Hybrid Approach** This hybrid design leverages complementary strengths: local UCT selection ensures principled sibling comparisons within subtrees, while global frontier selection mitigates UCT's myopia by allocating resources across subtrees. The approach achieves three key advantages: *(1) Computational efficiency* by eliminating redundant root-to-leaf traversals, *(2) Enhanced exploration coverage* by preventing the algorithm from getting trapped in locally promising but globally suboptimal subtrees, and *(3) Uncertainty-guided search* that leverages the policy's entropy to target regions expected to benefit from additional training supervision, with the bonus coefficient controlling the direction of this preference.

## 4 ADAPTIVE TRAINING STRATEGY WITH REPLAY BUFFER

While MCTS offers fine-grained credit assignment, applying it to every training example is computationally infeasible. To address this, we adopt an iterative filtering strategy with a replay buffer mechanism that focuses MCTS computation on challenging examples while preventing catastrophic forgetting of solved problems. The complete pipeline is depicted in Algorithm 1.

### 4.1 ITERATIVE TRAINING WITH PROGRESSIVE FILTERING

Our training process follows an iterative approach that progressively refines the training subset based on model performance. We begin by using the base RL model to perform an initial screening on the entire dataset $\mathcal{D}_{\text{hard}}$, creating the first training subset $\mathcal{D}_{\text{hard}}^{(0)}$ for MCTS-based RL training.

Specifically, the iterative training process proceeds as follows:

**Initial Subset Construction:** Given the base policy $\pi_{\theta^{(0)}}$, we evaluate its performance on the full training set $\mathcal{D}_{\text{train}}$ using direct rollouts and construct the initial hard subset:

$$\mathcal{D}_{\text{hard}}^{(0)} = \{x \in \mathcal{D}_{\text{train}} \mid \texttt{Pass1@K}(x, \pi_{\theta^{(0)}}) < \delta^{(0)}\}, \tag{8}$$

where $\texttt{Pass1@K}(x, \pi)$ represents the success rate when sampling $K = 4$ solutions for problem $x$ using policy $\pi$, and $\delta^{(0)} \in (0, 1)$ is the initial filtering threshold.

**Iterative Refinement:** After each training phase $i$, we re-evaluate the updated policy $\pi_{\theta^{(i)}}$ on the current hard subset and apply threshold-based filtering to create the next iteration's training set:

$$\mathcal{D}_{\text{hard}}^{(i+1)} = \{x \in \mathcal{D}_{\text{hard}}^{(i)} \mid \texttt{Pass1@K}(x, \pi_{\theta^{(i)}}) < \delta^{(i)}\}. \tag{9}$$

The filtering threshold $\delta^{(i)}$ is typically set to 25%, ensuring that only problems with insufficient success rates remain in the active training set. This progressive filtering concentrates computational resources on increasingly challenging problems as the model improves.

## 4.2 Replay Buffer with Cached Solutions

To prevent catastrophic forgetting and efficiently leverage previously discovered solutions, we maintain a replay buffer $\mathcal{R}$ that stores correct reasoning trajectories from earlier training phases.

**Buffer Population.** During each training iteration $i$, we identify problems that obtained correct solutions through MCTS rollouts but still fail to meet the filtering threshold after training:

$$\mathcal{R}_{\text{candidates}}^{(i)} = \{(x, \mathbf{t}_{\text{correct}}) \mid x \in \mathcal{D}_{\text{hard}}^{(i)}, \exists \mathbf{t}_{\text{correct}} \in \mathbb{T}(x), \texttt{Pass1@K}(x, \pi_{\theta^{(i)}}) < \delta^{(i)}\}. \tag{10}$$

These candidate trajectories are added to the replay buffer, attaining $\mathcal{R}^{(i+1)} = \mathcal{R}^{(i)} \cup \mathcal{R}_{\text{candidates}}^{(i)}$.

**Cached Solution Usage.** Instead of randomly sampling from the replay buffer, we employ a deterministic strategy that directly utilizes cached solutions when available. For each problem $x$ in the current training iteration, we first check whether a correct solution has been previously cached. This approach eliminates redundant MCTS computation for problems with known solutions while directing computational resources toward truly challenging, unsolved problems.

**Hybrid Rollout Strategy.** When processing problems in the current hard subset $\mathcal{D}_{\text{hard}}^{(i)}$, we apply different rollout strategies based on cache availability:

$$\text{Rollout}(x) = \begin{cases} \mathbf{t}_{\text{cached}} \cup \text{DirectRollouts}(x, \beta) & \text{if } (x, \mathbf{t}_{\text{cached}}) \in \mathcal{R}^{(i)}, \\ \text{MCTS}_{\text{full}}(x) & \text{otherwise.} \end{cases} \tag{11}$$

For problems with cached solutions, we directly incorporate the stored correct trajectory $\mathbf{t}_{\text{cached}}$ and supplement it with $\text{DirectRollouts}(x, \beta)$, which samples $\beta \cdot B$ additional solution attempts from the current policy $\pi_\theta(\cdot|x)$, where $\beta \in [0, 1]$ is implicitly determined by the number of cached solutions per problem (we allocate fewer DirectRollouts as more correct trajectories are cached) and $B$ is the standard sampling budget. For problems without cached solutions, we apply the complete MCTS search process $\text{MCTS}_{\text{full}}(x)$. Moreover, among the incorrect samples, we remove data containing garbled text or infinite repetitions. Based on empirical evidence, optimizing policies on such problematic data frequently leads to training collapse (Bai et al., 2025). The training dataset for each iteration is then constructed as:

$$\mathcal{T}_{\text{train}}^{(i)} = \underbrace{\bigcup_{x:(x,\mathbf{t}_{\text{cached}}) \in \mathcal{R}^{(i)}} \{\mathbf{t}_{\text{cached}} \cup \text{DirectRollouts}(x, \beta)\}}_{\text{Cached problems}} \cup \underbrace{\bigcup_{x:(x,\mathbf{t}_{\text{cached}}) \notin \mathcal{R}^{(i)}} \text{MCTS}_{\text{full}}(x)}_{\text{Unsolved problems}}. \tag{12}$$

This eliminates the need for artificial sampling ratios or complex batch composition strategies, as training data naturally incorporates both preserved knowledge and fresh exploration, tailored to problem-specific requirements. This has three key benefits: *(1) Computational efficiency* by avoiding redundant MCTS computation, *(2) Solution preservation* by guaranteeing the inclusion of cached correct trajectories, and *(3) Continued exploration* at minimal computational cost.

## 4.3 Tree-GRPO Training Objective

After constructing a search tree $\mathcal{T}$ for a sample question $x$ in the dataset $\mathcal{D}_{\text{train}}$, we develop our Tree-GRPO training objective. This objective combines q-value regularization with policy optimization to learn effectively from tree-structured reasoning traces.

**Q-Value Soft Clipping.** To address the q-value explosion problem for intermediate nodes while preserving meaningful gradients, we first apply *soft clipping* using the hyperbolic tangent function:

$$q(s_j) = \tanh\left(q^{(k_{\max})}(s_j)/\epsilon_q\right) \cdot q_{\max} \quad \text{for all } s_j \in \mathcal{T} \setminus \mathcal{S}_{\text{end}} \tag{13}$$

where $k_{\max}$ is the maximum rollout iterations, $\epsilon_q = 1.0$ is the temperature parameter, and $q_{\max} = 1$ defines the maximum allowable q-value magnitude.

This soft clipping approach prevents q-value explosion by maintaining all intermediate node q-values within $[-q_{\max}, q_{\max}]$, while offering several key advantages: *(i)* it naturally bounds q-values without hard discontinuities, *(ii)* it preserves gradients everywhere, preventing the zero-gradient problem that occurs with hard clipping when all values hit the same bound, and *(iii)* it maintains the relative ordering of q-values while compressing extreme outliers. Terminal node q-values remain unchanged as defined in Eq. 4. In practice, fewer than about 5% of intermediate node q-values fall into the saturation region and less than 0.5% lie near the boundaries, so clipping mainly compresses pathological tails rather than altering the bulk of the distribution.

**Training Objective.** With regularized q-values, we formulate and maximize the Tree-GRPO objective as:

$$
\mathcal{J}(\theta) = \mathbb{E}_{\mathbb{T} \sim \mathcal{T}, \mathbf{t}^i \sim \mathbb{T}, (s_j, o_j) \sim \mathbf{t}^i} \frac{1}{|s_j|} \sum_{k=1}^{|s_j|} \min \left( \rho_{j,k}(\theta) \hat{A}_{j,k}, \text{clip} \left( \rho_{j,k}(\theta), 1 - \epsilon_{\text{low}}, 1 + \epsilon_{\text{high}} \right) \hat{A}_{j,k} \right)
$$
(14)

where $\rho_{j,k}(\theta) = \frac{\pi_\theta(a_{j,k}|o_j, a_{j,<k})}{\pi_{\theta_{\text{old}}}(a_{j,k}|o_j, a_{j,<k})}$ is the importance ratio. The parameters $\epsilon_{\text{high}}$ and $\epsilon_{\text{low}}$ follow the Clip-Higher strategy of DAPO (Yu et al., 2025), while we also remove the KL regularization term $\mathbb{D}_{\text{KL}}$ to naturally diverge (Luo et al., 2025a; He et al., 2025a). An overlong buffer penalty is applied to responses that exceed a predefined maximum of 4096. The advantage function for node $s_j$ in trajectory $\mathbf{t}_i$ is computed using *sequence-level normalization* (Chu et al., 2025):

$$
\hat{A}_{j,k} = q(s_j) - \mu_{\mathbf{t}},
$$
(15)

where $\mu_{\mathbf{t}}$ is the average reward of the terminal nodes $\mathcal{S}_{\text{end}}$ throughout the tree $\mathbb{T}$. This normalization is crucial in practice, particularly for mitigating uncontrolled growth in response length. Notably, Tree-GRPO can be degraded to the vanilla DAPO if we consistently leverage the outcome reward $q(s_{\text{end}})$ as $q(s_j)$ for all intermediate nodes.

# 5 EXPERIMENTS

## 5.1 BENCHMARK PERFORMANCE EVALUATION

**Datasets and Base Models.** We train DeepSearch based on Nemotron-Research-Reasoning-Qwen-1.5B v2 (Liu et al., 2025a) and employ DeepMath-103K (He et al., 2025c) as the raw dataset. DeepMath-103K is a large-scale mathematical dataset designed to be highly challenging and rigorously decontaminated across numerous benchmarks. We evaluate DeepSearch against state-of-the-art 1.5B reasoning models on six mathematical benchmarks: AIME 2024/2025, AMC2023, MATH500 (Hendrycks et al., 2021), Minerva (Lewkowycz et al., 2022), and Olympiad (He et al., 2024). More experimental details are described in Appendix A.

**Baselines.** We compare against recent 1.5B models spanning different mechanisms: base models (Qwen2.5-Math variants), RL-trained models (DeepSeek-R1-Distill, STILL-3 (Team, 2025), Open-RS series (Dang & Ngo, 2025), advanced RL methods (DeepScaleR (Luo et al., 2025b), Nemotron variants), and search-based approaches (Qwen2.5-Math-Oat-Zero (Liu et al., 2025b)). Our evaluation methods and results are consistent with Hochlehnert et al. (2025).

**Results.** Table 1 shows DeepSearch-1.5B achieves 62.95% average accuracy, outperforming all baselines, including the previous best Nemotron-Research-Reasoning-Qwen-1.5B v2 (61.70%). DeepSearch-1.5B demonstrates consistent improvements across all benchmarks, with notable gains on AIME 2024 (53.65% vs 51.77%) and AMC (90.39% vs 88.83%). The 1.25 percentage-point improvement over the previous state of the art validates the effectiveness of integrating structured search into RLVR training rather than restricting it to inference-only use.

## 5.2 TRAINING EFFICIENCY ANALYSIS

To evaluate the practical viability of DeepSearch, we compare computational costs with extended training approaches that scale solely with the number of additional training steps. As shown in Table 2, extended training exhibits diminishing returns: 325 additional steps achieve 61.78% accuracy

Table 1: Performance comparison of 1.5B-scale language models on standard mathematical reasoning benchmarks. We report `Pass1@1` accuracy estimated with $n = 32$ samples. The best-performing results are highlighted in bold. All evaluations were conducted on a 128×H100 96G cluster.

| Model | AIME24 | AIME25 | AMC23 | MATH | Minerva | Olympiad | Avg |
|---|---|---|---|---|---|---|---|
| Qwen2.5-Math-1.5B | 8.33 | 6.35 | 44.06 | 66.67 | 18.42 | 30.74 | 29.10 |
| Qwen2.5-Math-1.5B-Instruct | 10.10 | 8.85 | 55.08 | 74.83 | 29.32 | 40.00 | 36.37 |
| DeepSeek-R1-Distill-Qwen-1.5B | 31.15 | 24.06 | 72.81 | 85.01 | 32.18 | 51.55 | 49.46 |
| STILL-3-1.5B | 31.46 | 25.00 | 75.08 | 86.24 | 32.77 | 53.84 | 50.73 |
| Qwen2.5-Math-1.5B-Oat-Zero | 20.00 | 10.00 | 52.50 | 74.20 | 26.84 | 37.78 | 36.89 |
| Open-RS1-1.5B | 30.94 | 22.60 | 73.05 | 84.90 | 29.92 | 52.82 | 49.04 |
| Open-RS2-1.5B | 28.96 | 24.37 | 73.52 | 85.06 | 29.74 | 52.63 | 49.05 |
| Open-RS3-1.5B | 30.94 | 24.79 | 72.50 | 84.47 | 29.11 | 52.25 | 49.01 |
| DeepScaleR-1.5B | 38.54 | 30.52 | 80.86 | 88.79 | 36.19 | 58.95 | 55.64 |
| Nemotron-Research-Reasoning-Qwen-1.5B v1 | 45.62 | 33.85 | 85.70 | 92.01 | 39.27 | 64.56 | 60.17 |
| Nemotron-Research-Reasoning-Qwen-1.5B v2 | 51.77 | 32.92 | 88.83 | 92.24 | 39.75 | 64.69 | 61.70 |
| DeepSearch-1.5B | **53.65** | **35.42** | **90.39** | **92.53** | **40.00** | **65.72** | **62.95** |

Table 2: Comparison of methods on efficiency and performance, which are trained from DeepSeek-R1-Distill-Qwen-1.5B.

| Method | RLVR | Steps | Samples (K) | Time (h) | GPU Hours | Math Score |
|---|---|---|---|---|---|---|
| DeepSeek-R1-Distill-Qwen-1.5B | – | – | – | – | – | 49.46 |
| Nemotron-Research-Reasoning-Qwen-1.5B v1 | DAPO | 2000 | – | – | 16000 | 60.10 |
| Nemotron-Research-Reasoning-Qwen-1.5B v2 | DAPO | 3000 | – | – | 24000 | 61.70 |
| Extended Training | DAPO | +325 | 665.6 | 20.4 | 326.4 | 61.78 |
| Extended Training | DAPO + KL | +785 | 1607.7 | 49.3 | 788.8 | 62.08 |
| Extended Training | DAPO + KL | +1875 | 3840.0 | 117.7 | 1883.2 | 62.02 |
| DeepSearch-1.5B | Tree-GRPO | +50 | 102.4 | 20.6 | 330 | **62.95** |

using 326.4 GPU hours, whereas 1,875 steps plateau at 62.02% accuracy despite consuming 1,883.2 GPU hours. This reveals the fundamental limitation of depth-first scaling, where performance gains become marginal as computational investment grows exponentially.

DeepSearch achieves superior results through algorithmic innovation rather than brute-force computation. With only 50 additional training steps, DeepSearch reaches 62.95% accuracy using 330 GPU hours—outperforming the most extensive baseline (1,883.2 hours) while using 5.7× fewer resources. This efficiency stems from a structured search that extracts maximum value from each training step by systematically exploring diverse solution paths.

Figure 2 illustrates the training dynamics over 20 hours following 3K RLVR training. DAPO exhibits gradual linear improvement with a shallow slope, while DeepSearch demonstrates more efficient learning through structured exploration. The superior convergence properties suggest that RLVR bottlenecks stem from the quality of exploration rather than from insufficient training time.

These results challenge the common assumption that improving RLVR performance requires proportional increases in computational budget. Compared with the training of Nemotron-Research-Reasoning-Qwen-1.5B v2, DeepSearch-1.5B achieves a 72× efficiency improvement, highlighting a qualitative shift toward algorithmically driven optimization rather than resource-driven scaling.

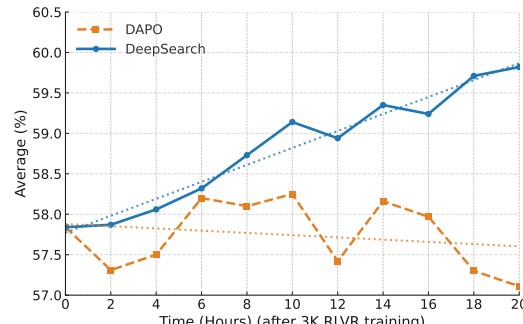

Figure 2: Average performance (AIME 2024, AIME 2025, and AMC 2023) of **DAPO** and **DeepSearch** after 3K RLVR training. Markers denote evaluations, while dotted lines indicate linear trends.

This suggests that systematic exploration can offer a more practical and scalable path for advancing RLVR capabilities than extended high-compute training.

Table 3: Ablation study of different search strategies in DeepSearch. We compare vanilla UCT with our proposed global frontier selection under varying depth bonus functions $D(d(s))$. Reported metrics include search statistics such as average search depth, trajectory entropy, and trajectory reward, as well as computational cost, measured by the number of iterations, average per-iteration time (in seconds), and per-tree time (in seconds). Results are presented as mean $\pm$ standard deviation.

| Method | $D(d(s))$ | Search Metrics | | | Computational Cost | | |
| --- | --- | --- | --- | --- | --- | --- | --- |
| | | Depth | Entropy | Reward | Num. Iter. | Time Per Iter. | Time Per Tree |
| Vanilla UCT | – | $20.11 \pm 4.72$ | $1.23 \pm 0.29$ | $-0.82 \pm 0.57$ | $209.6 \pm 14.8$ | $5.63 \pm 0.21$ | $1179.6 \pm 95.0$ |
| **Global Frontier Selection** | | | | | | | |
| $\lambda_1 = 0.4$ | – | $20.28 \pm 4.80$ | $1.23 \pm 0.29$ | $-0.65 \pm 0.76$ | $187.7 \pm 16.2$ | $5.76 \pm 0.19$ | $1087.7 \pm 105.0$ |
| $\lambda_1 = 0.4, \lambda_3 = 0.01$ | $\log(d(s)+1)$ | $20.33 \pm 4.77$ | $1.23 \pm 0.30$ | $-0.65 \pm 0.76$ | $185.5 \pm 15.9$ | $5.85 \pm 0.19$ | $1080.3 \pm 102.2$ |
| $\lambda_1 = 0.4, \lambda_3 = 0.01$ | $d(s)$ | $21.55 \pm 5.13$ | $1.24 \pm 0.29$ | $-0.76 \pm 0.65$ | $85.7 \pm 7.7$ | $5.61 \pm 0.12$ | $480.9 \pm 41.9$ |
| $\lambda_1 = 0.4, \lambda_2 = 0.4, \lambda_3 = 0.01$ | $\sqrt{d(s)/d_\mathcal{T}}$ | $20.83 \pm 4.71$ | $1.31 \pm 0.30$ | $-0.79 \pm 0.62$ | $92.5 \pm 22.5$ | $5.48 \pm 0.13$ | $505.2 \pm 114.8$ |
| **$\lambda_1 = 0.4, \lambda_3 = 0.01$** | $\sqrt{d(s)/d_\mathcal{T}}$ | **$20.29 \pm 4.83$** | **$1.24 \pm 0.29$** | **$-0.65 \pm 0.76$** | **$189.3 \pm 14.7$** | **$5.66 \pm 0.14$** | **$1070.7 \pm 87.3$** |

## 5.3 SEARCH STRATEGY ABLATION

Table 3 compares our global frontier selection against vanilla UCT under different configurations on 1.2K samples from extremely hard DeepMath-103K problems.

**Global vs. Local Selection.** Our global frontier selection ($\lambda_1 = 0.4$) reduces iterations by 10.4% ($209.6 \rightarrow 187.7$) and improves trajectory rewards ($-0.82 \rightarrow -0.65$) compared to vanilla UCT, while maintaining similar search depth and entropy. This demonstrates that comparing frontier nodes across the entire tree is more efficient than traditional root-to-leaf UCT traversals.

**Depth Bonus Impact.** We evaluate three depth bonus functions $D(d(s))$: (i) Logarithmic $\log(d(s) + 1)$ provides minimal improvements, (ii) Linear $d(s)$ achieves the most aggressive efficiency gains with 59% reduction in per-tree time ($1179.6s \rightarrow 480.9s$) and deepest exploration (21.55 depth), but at cost of solution quality ( -0.76 reward), (iii) Square root $\sqrt{d(s)/d_T}$ offers the best balance, maintaining search quality (-0.65 reward) with significant computational savings.

**Uncertainty Bonus.** Adding uncertainty weighting ($\lambda_2 = 0.4$) increases exploration diversity (entropy $1.23 \rightarrow 1.31$) by prioritizing high-uncertainty policy regions, but introduces computational variability ($92.5 \pm 22.5$ iterations).

**Configuration Selection.** We adopt $\sqrt{d(s)/d_T}$ with $\lambda_1 = 0.4, \lambda_3 = 0.01$ as our default, balancing computational efficiency (189.3 iterations), search quality (-0.65 reward), and stable performance. This configuration eliminates UCT's redundant traversals while maintaining principled exploration through quality potential and depth guidance.

## 5.4 ALGORITHM EVOLUTION AND COMPONENT CONTRIBUTIONS

To understand the individual contributions of each component, we present a systematic ablation study of our DeepSearch algorithm in Table 4. Starting from the `Nemotron-Qwen-1.5B v2` base, we incrementally add components and analyze their impact:

(i) **Vanilla DeepSearch Foundation.** We begin with a basic MCTS integration using a simple q-value update rule:

$$q^{(m)}(s_i) = \begin{cases} q^{(m-1)}(s_i) + \gamma(i,l) \cdot q^{(m)}(s_{\text{end}}) & \text{if } q^{(m-1)}(s_i) \cdot q^{(m)}(s_{\text{end}}) \geq 0, \\ \max\bigl(q^{(m-1)}(s_i) + \gamma(i,l) \cdot q^{(m)}(s_{\text{end}}), 0\bigr) & \text{otherwise.} \end{cases}$$

This assigns constant values to nodes along correct reasoning paths but yields only limited improvement over the baseline. (ii) **Enhanced Q-Value Updates with Outcome Rewards.** We replace the simple update with our constrained backup rule (Eq. 5) and use outcome-based advantages $\hat{A}_{j,k} = q(s_{\text{end}})$ for all nodes. This provides more stable credit assignment and yields meaningful improvements. (iii) **Fine-Grained Node-Level Advantages.** Moving beyond outcome-only rewards, we assign node-specific advantages $\hat{A}_{j,k} = q(s_j)$ based on each node's individual q-value. This enables more precise credit assignment across different reasoning steps. (iv) **Standard Advantage Normalization.** We implement standard normalization as $\hat{A}_{j,k} = \frac{q(s_j) - \mu_{\mathbf{t}}}{\sigma_{\mathbf{t}} + \varepsilon}$, where $\sigma_{\mathbf{t}}$ is the standard deviation of the rewards of the terminal nodes $\mathcal{S}_{\text{end}}$ throughout the tree $\mathbb{T}$. The constant $\varepsilon$ prevents numerical instability when the variance is small. This stabilizes training but introduces variance-based scaling. (v) **Mean-Only Normalization.** We adopt mean-only normalization (Eq. 15). This

Table 4: Ablation study illustrating the step-by-step evolution of **DeepSearch**. Starting from Vanilla DeepSearch with a simple $q$-update, we progressively add outcome-reward–based and fine-grained advantages, standard or mean-only normalization, and frontier node selection.

| Model / Change | AIME24 | AIME25 | AMC23 | MATH | Minerva | Olympiad | Avg |
|---|---|---|---|---|---|---|---|
| Nemotron-Research-Reasoning-Qwen-1.5B v2 | 51.77 | 32.92 | 88.83 | 92.24 | 39.75 | 64.69 | 61.70 |
| + Vanilla DeepSearch | 51.98 | 34.06 | 86.64 | 87.00 | 37.96 | 64.00 | 60.27 |
| + New $q$ Update & Coarse-grained Token Scores | 51.04 | **35.73** | 86.48 | 90.66 | 39.14 | 65.23 | 61.38 |
| + New $q$ Update & Fine-grained Token Scores | 50.52 | 35.52 | 88.83 | 91.70 | 39.71 | 64.81 | 61.85 |
| + Standard Advantages Normalization | 52.60 | 35.00 | 89.30 | 92.44 | 39.29 | 64.99 | 62.27 |
| + Mean-only Advantages Normalization | 51.98 | **35.73** | 89.06 | 91.88 | 39.58 | 65.71 | 62.32 |
| + Frontier Selection | **53.65** | 35.42 | **90.39** | **92.53** | **40.00** | **65.72** | **62.95** |

Table 5: **Ablation on different trajectory selection strategies.** We compare using the most confident incorrect trajectory, a randomly selected incorrect trajectory, and the least confident incorrect trajectory. Confidence is measured via average trajectory entropy.

| Method | AIME24 | AIME25 | AMC23 | MATH | Minerva | Olympiad | Avg |
|---|---|---|---|---|---|---|---|
| Nemotron-Research-Reasoning-Qwen-1.5B v2 | 51.77 | 32.92 | 88.83 | 92.24 | 39.75 | 64.69 | 61.70 |
| + Random incorrect trajectory | 52.41 | 34.23 | 89.31 | 92.21 | 39.97 | 64.41 | 62.09 |
| + Least confident incorrect trajectory | 52.16 | 33.77 | 88.68 | 91.96 | 40.11 | 64.73 | 61.90 |
| + **Most confident incorrect trajectory** | **53.65** | **35.42** | **90.39** | **92.53** | **40.00** | **65.72** | **62.95** |

addresses miscalibration issues in GRPO while maintaining stable advantage scaling. (Bereket & Leskovec, 2025). (vi) **Global Frontier Selection.** Finally, we integrate our novel frontier selection strategy (Eq. 6), which prioritizes promising expansion candidates across the entire search tree rather than following traditional root-to-leaf UCT-like traversals.

Moreover, to justify our choice of prioritizing the *most confident* incorrect trajectory when no correct solution is found, we compare three selection strategies in Tab. 5. Selecting the trajectory with the lowest average entropy, representing the model's *most confident* mistake, achieves the best average performance (+0.86 over random and +1.05 over least confident), and is the only method that consistently improves results across all benchmarks. This confirms that high-confidence incorrect trajectories expose *systematic reasoning errors* that benefit most from targeted supervision, whereas random or low-confidence errors tend to be noisy or uninformative.

The results demonstrate that each component contributes meaningfully to the final performance, with frontier selection providing the largest single improvement. The cumulative effect shows that systematic exploration and fine-grained credit assignment are essential for maximizing the benefits of search-augmented RLVR.

### 5.5 Discussion: Why Tree-Based Exploration Helps

Direct rollouts from $\pi_\theta$ behave like blind sampling: they quickly collapse into high-probability but low-diversity regions, rarely reaching deeper reasoning paths. As a result, they fail to uncover the informative trajectories required to correct systematic reasoning errors. In contrast, MCTS provides *structured* exploration. Instead of distributing compute uniformly at random, it allocates search effort according to frontier quality, uncertainty, and depth. This yields three intuitive benefits:

(1) **Promising branches are expanded first.** Frontier scoring prioritizes nodes with high value potential, preventing premature collapse into shallow modes. (2) **Overconfident mistakes become learnable.** When no correct solution exists, we select the lowest-entropy incorrect trajectory, exposing systematic reasoning failures that direct rollouts rarely sample. (3) **Deep reasoning chains are reachable.** Depth bonuses encourage exploration of multi-step trajectories that direct rollouts fail to reach due to early-token drift.

### 6 Conclusion

We introduced DeepSearch to integrate tree-based search into RLVR training, thereby addressing exploration bottlenecks that lead to performance plateaus. It achieves an average accuracy of 62.95% on mathematical reasoning benchmarks, establishing a new state-of-the-art for 1.5B models while using 5.7× fewer GPU hours. This suggests that systematic exploration during training can, in many cases, provide greater gains than simply extending training time, highlighting the value of scaling training breadth through algorithmic innovation rather than relying solely on deeper compute.

ACKNOWLEDGMENT

This work was supported in part by RS-2024-00457882, National AI Research Lab Project.

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

# A    EXPERIMENTAL DETAILS

This section provides comprehensive details of our experimental setup, including system implementation, training configurations, MCTS parameters, optimization strategies, and evaluation protocols used in our DeepSearch framework.

## A.1    TRAINING DATA AND CONFIGURATION

We implement our DeepSearch system using the veRL framework (Sheng et al., 2024). Training is conducted on 16×H100 GPUs, each with 96GB of memory; evaluation is run on a larger 128×H100 cluster to reduce wall-clock time. The policy model is initialized with Nemotron-Research-Reasoning-Qwen-1.5B v2 (Liu et al., 2025a) (updated July 23rd). To ensure a fair comparison with a well-aligned policy, we additionally perform DAPO-based extended training using the Nemotron-Research-Reasoning-Qwen-1.5B-v2 initialization, with the same training configuration as DeepSearch.

Our training methodology uses the DeepMath-103K (He et al., 2025c) dataset as $D_{\text{train}}$ and employs a DeepScaleR-style prompt template that instructs the model to "Let's think step by step and output the final answer within \boxed{}." To manage computational constraints, we truncate the left side, keeping the most recent tokens with a maximum prompt length of 2,048 tokens and limiting response generation to 16,384 tokens. The training process utilizes a global batch size of 256 samples, implemented through the DAPO-style Dynamic Batching strategy (Yu et al., 2025) to optimize memory utilization and training efficiency.

## A.2    MONTE CARLO TREE SEARCH IMPLEMENTATION

Our MCTS implementation incorporates several strategic design choices to balance search efficiency and solution quality. The exploration coefficient ($\lambda$) for UCT Local Selection is set to 2.0, providing an optimal exploration-exploitation trade-off for mathematical reasoning tasks. The search architecture operates with a maximum depth of 64 levels, with each node allocated 256 tokens and expanding 8 children during the expansion phase. For entropy-based selection, we estimate average trajectory entropy using only tokens that appear in the response, rather than the entire per-position vocabulary, for computational efficiency. This average token negative log-probability along the sampled trajectory serves as a proxy for decision confidence.

To enhance search effectiveness, the system employs Global Frontier Selection for backtracking and applies a square-root function to depth-based bonuses, encouraging deeper exploration when beneficial. The global $\lambda_3$ parameter is configured to 0.01 for our frontier priority scoring, while an overlong buffer of 4,096 tokens with a penalty factor of 1.0 accommodates lengthy reasoning chains typical in complex mathematical problems.

## A.3    ADVANTAGE ESTIMATION AND OPTIMIZATION

For advantage estimation, we implement the Grouped Relative Policy Optimization (GRPO) (Shao et al., 2024) estimator with sibling mean normalization to ensure stable learning dynamics. The Q-value soft clipping mechanism operates at a temperature of 1.0 with the maximum q-value magnitude set to 1.0, while incomplete trajectories receive a penalty score of $-1.0$ to discourage premature termination. Standard deviation normalization is disabled to prevent numerical instability during training.

The actor model is optimized using AdamW with a conservative learning rate of $1 \times 10^{-6}$ and 10 warmup steps, along with weight decay of 0.1 and gradient clipping at 1.0 to stabilize convergence. We follow the Clip-Higher strategy in DAPO (Yu et al., 2025) and set the lower and upper clipping thresholds to 0.2 and 0.28, respectively, with a clipping ratio of 10.0. Training proceeds with mini-batches of 32 samples per policy update using token-mean loss aggregation, while dynamic batch sizing accommodates up to 18,432 tokens per GPU. The entropy coefficient is set to 0 for pure exploitation, and the KL divergence loss is disabled to maximize performance on the target mathematical reasoning tasks.

### A.4   SAMPLING AND REWARD CONFIGURATION

During rollout generation, we set the sampling temperature to 1.0 and $top\_p$ to 1.0, and disable $top\_k$ filtering to maintain diverse response generation. The system generates 8 rollouts per prompt, aligning with the expansion width parameter, within a context length of 18,432 tokens. This configuration ensures comprehensive exploration of the solution space while maintaining computational feasibility. During evaluation, we uniformly use a low temperature of 0.6 and $top\_p$ of 0.95.

Our reward system implements a custom mathematical scoring function based on (`compute_score`) from the `math_dapo.py` module, designed to evaluate mathematical reasoning accuracy. We extract the final boxed answer by locating the last occurrence of \boxed{} in the trajectory and apply the same text-normalization logic as veRL's DAPO recipe to both prediction and ground-truth. The reward mechanism handles responses up to 16,384 tokens, following ProRL (Liu et al., 2025a) and ensuring consistent evaluation across varying response lengths.

### A.5   TRAINING PROTOCOL

The complete training protocol spans 100 steps with model checkpointing performed every 5 steps. This frequent checkpointing strategy ensures robust model preservation and enables detailed analysis of learning progress throughout training.

## B   JUSTIFICATION OF DEEPSEARCH DESIGN CHOICE

### B.1   MAXIMUM DEPTH AND EXPANSION LENGTH

**Data-driven determination of maximum search depth.**   We set the maximum search depth through two jointly defined factors: a rollout budget of 16,384 tokens and an expansion length of 256 tokens per node, yielding an approximate upper bound of $16,384/256 \approx 64$ expansion steps. In practice, many trajectories terminate earlier, so this acts as an upper bound rather than a hard requirement. Most importantly, this limit is derived from empirical statistics of the reasoning-length distribution in DeepMath-103K, which confirms that 16,384 tokens comfortably cover full reasoning chains for nearly all problems. The cap, therefore, reflects a principled trade-off between modeling requirements and training efficiency—not an arbitrary constant.

**Empirical validation of 256-token expansion length.**   We evaluate several expansion lengths (128, 256, 512, 1024 tokens) and observe a clear trade-off:

- **Search-space exploration:** Longer expansions reduce tree breadth and make MCTS less likely to discover correct solution paths.
- **Computational cost:** Shorter expansions significantly increase prefix re-encoding ("pre-fill") costs, which dominate GPU time in MCTS exploration.

The 256-token step size consistently gives the best balance between coverage, training stability, and computational efficiency across all experiments.

**Summary.**   Both design choices of the depth cap and the expansion length arise from data-driven analysis and controlled comparisons. Together, they ensure that DeepSearch maintains wide search coverage while avoiding unnecessary computational overhead.

### B.2   RATIONALE BEHIND THE ASYMMETRIC Q-VALUE BACKUP RULE

Our constrained backup rule in Eq. 5 maintains for every intermediate node a scalar q-value representing its estimated tendency to lead toward a correct solution under the current policy and search process. When a terminal trajectory returns a reward signal ($+1$ for correct, $-1$ for incorrect or incomplete), the update is *asymmetric*, depending on the relationship between the node's current q-value and the reward.

**Same-sign reinforcement.**   When a node's existing q-value and the terminal reward share the same sign, the update simply adds the depth-decayed reward. Thus, nodes that frequently appear on correct trajectories naturally accumulate stronger positive evidence, while nodes that are consistently involved in incorrect trajectories accumulate stronger negative evidence.

**Negative-to-positive transitions.**   If a node previously held a negative q-value—indicating it only appeared in incorrect trajectories—but it now lies on a correct trajectory, we discard the accumulated negative value and overwrite it with the decayed positive reward. Intuitively, once a node is ever shown to lead to a valid solution, we reclassify it as potentially valuable rather than penalizing it for past failures.

**Positive-to-negative suppression.**   If a node already has a positive q-value, meaning it has appeared at least once in a successful reasoning path, we do *not* propagate new negative signals to it. A node proven capable of contributing to a correct solution should not be downgraded due to occasional failed attempts, which could arise from stochastic expansions or unrelated branches.

**Guaranteed invariants.**   This asymmetric update mechanism enforces two key invariants:

1. Any node that has ever appeared on a correct trajectory retains a non-negative q-value.
2. Only nodes that have *never* observed on a correct trajectory are allowed to accumulate stable negative values.

Together, these invariants ensure that the search process robustly separates "potentially useful" nodes from those that consistently lead to dead ends, thereby stabilizing MCTS exploration and preventing destructive interference from noisy, incorrect rollouts.

### B.3   Evolution of Cached vs. Unsolved Problems During Adaptive Training

To quantify how adaptive training reallocates computation over time, we track the ratio of cached problems (those with at least one verified correct trajectory discovered by MCTS) to unsolved problems (those with no verified correct trajectory). Because Round 1 begins with an empty replay buffer, the cached proportion is initially 0:1. As training progresses, MCTS increasingly cracks problems in the hard subset, causing cached solutions to accumulate round by round.

Table 6: Number of cached vs. unsolved problems during the 5-round adaptive training.

| Round | # Cached | # Unsolved | Cached Ratio |
|:-----:|:--------:|:----------:|:------------:|
| 1 | 0 | 13,658 | 0% |
| 2 | 765 | 12,893 | 5.6% |
| 3 | 1,452 | 9,423 | 13.4% |
| 4 | 2,243 | 7,423 | 23.2% |
| 5 | 2,894 | 5,829 | 33.2% |

The monotonic increase in cached solutions demonstrates that DeepSearch progressively transfers problems from the unsolved pool into the replay buffer as MCTS uncovers verifiable solution paths. By Round 5, one-third of the hard subset has at least one cached solution, while the unsolved set contracts to a concentrated collection of extremely difficult tail cases. This behavior validates our description in Sec. 4.2: adaptive training naturally shifts computational effort toward the most challenging remaining problems, enabling more efficient use of MCTS and contributing to the strong sample efficiency shown in Table 2 and Figure 2.

### B.4   On the Choice of a Fixed Filtering Threshold $\delta$

The filtering threshold $\delta = 25\%$ used in our adaptive training pipeline is a simple yet empirically effective heuristic. At each round, we compute each problem's success rate using `Pass1@K` with $K = 4$ direct rollouts and retain only those samples whose success rate falls below $\delta$ for inclusion in the next hard set. This mechanism filters out problems the model has effectively mastered while preserving sufficiently challenging cases that benefit most from training-time MCTS exploration.

Although one could design an adaptive $\delta$ schedule, such as one based on training progress or the evolving difficulty distribution, we intentionally avoid such complexity for two reasons. First, adaptive thresholding would introduce additional schedules and hyperparameters (e.g., functional forms across rounds and smoothing coefficients), complicating the design space and obscuring the specific contribution of MCTS-based exploration. Second, our primary aim is to demonstrate that *even under a fixed, straightforward hard-sample selection rule*, integrating structured tree search into RLVR training already yields stable and significant gains.

Thus, while adaptive thresholding remains a promising direction aligned with curriculum-style learning, we prioritize methodological simplicity and clear attribution of improvements in this work.

## C   COMPUTATION PROFILING

To better understand the computational characteristics of DeepSearch, we profile the per-tree runtime of our final configuration ($\lambda_1 = 0.4$, $\lambda_3 = 0.01$), with components ranked from smallest to largest cost in Tab. 7. The results show that all CPU-side operations—including prompt construction, terminal-state checking, rank synchronization, input preparation, node expansion, and output post-processing—collectively account for only a few milliseconds and are negligible relative to the total cost. In contrast, *GPU-based policy inference* overwhelmingly dominates the runtime, with $t_{\text{generate}} = 1060.76$ ms comprising more than 99% of the per-tree computation. This profiling confirms that the primary bottleneck lies in sequence generation rather than in MCTS logic, implying that further speedups will mainly come from optimizing model inference (e.g., batching, KV-cache reuse, speculative decoding) rather than from modifying the search algorithm's structure.

Table 7: **Computation time breakdown per MCTS tree** under the final search configuration ($\lambda_1 = 0.4$, $\lambda_3 = 0.01$). Components are ranked from smallest to largest cost. Times are reported in milliseconds as mean $\pm$ standard deviation.

| Component | Time |
|---|---|
| Output post-processing ($t_{\text{proc\_outputs}}$) | $0.01 \pm 0.00$ |
| Rank synchronization ($t_{\text{sync\_ranks}}$) | $0.07 \pm 0.01$ |
| Terminal-state checking ($t_{\text{check\_terminal}}$) | $0.09 \pm 0.02$ |
| Prompt construction ($t_{\text{get\_prompts}}$) | $0.38 \pm 0.08$ |
| Node expansion ($t_{\text{expand\_node}}$) | $0.96 \pm 0.18$ |
| Input preparation ($t_{\text{prep\_inputs}}$) | $1.89 \pm 0.36$ |
| **Policy inference / generation ($t_{\textbf{generate}}$)** | $1060.76 \pm 106.59$ |

## D   PSEUDOCODE OF DEEPSEARCH

Algorithm 1 presents the complete DeepSearch framework, integrating MCTS-based exploration with adaptive training and replay buffer management. The algorithm operates via iterative refinement, progressively allocating computational resources to challenging problems while preserving previously solved solutions through intelligent caching. This integrated approach focuses on training-time exploration, enabling models to learn from both correct solutions and systematic exploration processes rather than relying solely on outcome-based supervision.

---

**Algorithm 1** DeepSearch with Global Frontier Selection and Iterative Filtering

---

**Require:** Initial policy $\pi_{\theta^{(0)}}$, training set $\mathcal{D}_{\text{train}}$, verifier $\mathcal{V}$, filtering threshold $\delta$

1: Initialize $\mathcal{D}_{\text{hard}}^{(0)} \leftarrow \{x \in \mathcal{D}_{\text{train}} \mid \texttt{Pass1@K}(x, \pi_{\theta^{(0)}}) < \delta^{(0)}\}, \mathcal{R}^{(0)} = \emptyset$
2: **for** training iteration $i = 0, 1, 2, \ldots$ **do**
3:     Initialize training trajectories $\mathcal{T}_{\text{train}}^{(i)} \leftarrow \emptyset$
4:     **for** each batch $\mathcal{B}^{(i)} \in \mathcal{D}_{\text{hard}}^{(i)}$ **do**
5:         **for** each problem $x \in \mathcal{B}^{(i)}$ **do**
6:             **if** $(x, \mathbf{t}_{\text{cached}}) \in \mathcal{R}^{(i)}$ **then**                    ▷ Use cached solution
7:                 $\mathcal{T}_x \leftarrow \{\mathbf{t}_{\text{cached}}\} \cup \text{DirectRollouts}(x, \beta)$
8:                 $\mathcal{T}_{\text{train}}^{(i)} \leftarrow \mathcal{T}_{\text{train}}^{(i)} \cup \mathcal{T}_x$
9:             **else**                                 ▷ Apply full MCTS search
10:                 **MCTS Search:**
11:                 Initialize search tree $\mathcal{T}$ with root node $x$
12:                 **for** rollout iteration $k = 1, 2, \ldots$ **do**
13:                     **if** $k = 1$ **then**                 ▷ Initial expansion from root
14:                         Select root node $s^* = x$ for expansion
15:                     **else**
16:                         **Global Frontier Selection:**
17:                         Compute frontier set $\mathcal{F} = \{s \in \mathcal{T} \mid \xi(s) = 0, s \notin \mathcal{S}_{\text{end}}, d(s) < d_{\mathcal{T}}\}$
18:                         Compute frontier priority scores (Eq. 7)
19:                         Select node $s^* = \arg\max_{s \in \mathcal{F}} F(s)$ for expansion
20:                     **end if**
21:                   **Local Expansion with UCT Selection:**
22:                   Generate $n$ candidates $\{s_j\}_{j=1}^n \sim \pi_\theta(\cdot \mid o_{s^*})$ from $s^*$
23:                   Continue expansion until terminal nodes $\mathcal{S}_{\text{end}}^{(k)}$ are reached
24:                   **Evaluation with Entropy-based Guidance**
25:                   Partition: $\mathcal{S}_{\text{correct}}^{(k)} = \{s \in \mathcal{S}_{\text{end}}^{(k)} \mid \mathcal{V}(s) = 1\}, \mathcal{S}_{\text{incorrect}}^{(k)} = \{s \in \mathcal{S}_{\text{end}}^{(k)} \mid \mathcal{V}(s) = 0\}$
26:                   **if** $|\mathcal{S}_{\text{correct}}^{(k)}| \geq 1$ **then**
27:                       Extract trajectories $\mathbb{T}(x)$ from search tree $\mathcal{T}$
28:                       $\mathcal{T}_{\text{train}}^{(i)} \leftarrow \mathcal{T}_{\text{train}}^{(i)} \cup \mathbb{T}(x)$
29:                   **else**
30:                       Select most confident negative: $s_{\text{neg}}^* = \arg\min_{s \in \mathcal{S}_{\text{incorrect}}^{(k)}} \bar{H}(\mathbf{t}(s))$
31:                   **end if**
32:                   **Heuristic Score Backup:**
33:                   Select trajectory $\mathbf{t}^*$ (correct solution or $\mathbf{t}(s_{\text{neg}}^*)$)
34:                   Assign terminal rewards (Eq. 4)
35:                   **for** each node $s_j$ in $\mathbf{t}^*$ **do**
36:                     Update Q-values using constrained backup rule (Eq. 5)
37:                   **end for**
38:              **end for**
39:             **end if**
40:             **Replay Buffer Update:**
41:             **if** MCTS found correct solutions but $\texttt{Pass1@K}(x, \pi_{\theta^{(i)}}) < \delta^{(i)}$ **then**
42:                 Add $(x, \mathbf{t}_{\text{correct}})$ to $\mathcal{R}^{(i+1)}$ for any correct $\mathbf{t}_{\text{correct}} \in \mathbb{T}(x)$
43:             **end if**
44:         **end for**
45:         **Policy Update:**
46:         Update policy $\pi_{\theta^{(i+1)}}$ using Tree-GRPO objective on $\mathcal{T}_{\text{train}}^{(i)}$ (Eq. 13 and Eq. 14)
47:     **end for**
48:     Re-evaluate and filter: $\mathcal{D}_{\text{hard}}^{(i+1)} = \{x \in \mathcal{D}_{\text{hard}}^{(i)} \mid \texttt{Pass1@K}(x, \pi_{\theta^{(i+1)}}) < \delta^{(i+1)}\}$
49: **end for**

---

# E    QUALITATIVE CASE STUDIES

To complement the benchmark results in the main text, we provide 6 representative cases between our **DeepSearch-1.5B** and **Nemotron-Research-Reasoning-Qwen-1.5B v2** on the standard math reasoning benchmarks. Each case study is selected from a different dataset and illustrates typical differences in reasoning behavior, including hesitation patterns, error correction, and final answer reliability. For each problem, we show the original question and the model-generated reasoning traces. For brevity, we present lightly abridged traces with minor formatting edits, while preserving the essential reasoning.

Across these 6 examples, DeepSearch-1.5B consistently reaches the correct solution with shorter, more stable reasoning chains, while the Nemotron-Research-Reasoning-Qwen-1.5B v2 often falls into long "wait"-style hesitation loops, revisits the same partial calculations, and in several cases converges to an incorrect final answer despite extensive deliberation. For instance, in the symbolic-logic example from MATH, Nemotron-Research-Reasoning-Qwen-1.5B v2 emits over a thousand explicit hesitation markers, whereas DeepSearch-1.5B solves the problem with two orders of magnitude fewer such tokens while maintaining algebraic consistency and arriving at the correct value. This pattern holds uniformly across different types of problems, highlighting that training-time exploration not only reduces unnecessary verbal dithering but also improves robustness of multi-step mathematical reasoning.

---

**MATH** (*Symbol Logic / Constraint Satisfaction*)

**Question.** The symbols $\triangle$, $\square$, $\diamond$, $\clubsuit$ represent four different integers from 1 to 9. Using the equations below, what is the value of $\square$?

$$\triangle + \square = \clubsuit$$
$$\triangle + \triangle = \diamond + \diamond + \diamond + \diamond + \diamond$$
$$\triangle + \triangle = \clubsuit + \diamond.$$

**DeepSearch-1.5B.**

> Okay, so I have this problem where different symbols represent different integers from 1 to 9. The symbols are triangle ($\triangle$), square ($\square$), diamond ($\diamond$), and clubsuit ($\clubsuit$). The given equations are:
>
> 1. $\triangle + \square = \clubsuit$
> 2. $\triangle + \triangle = \diamond + \diamond + \diamond + \diamond + \diamond$
> 3. $\triangle + \triangle = \clubsuit + \diamond$
>
> And I need to find the value of $\square$. Let's write these equations more clearly.
>
> [...]
> ng $a = 5$ and $c = 2$: $10 = d + 2 \implies d = 8$
> From equation 1: $a + b = d \implies 5 + b = 8 \implies b = 3$.
> Thus, the values are: - $a = 5$ - $b = 3$ - $c = 2$ - $d = 8$
> All values are distinct integers from 1 to 9. Therefore, the value of $\square$ (which is $b$) is 3.
> The value of $\square$ **is** $\boxed{3}$.

**Nemotron-Research-Reasoning-Qwen-1.5B v2.**

> Okay, so I need to find the value of $\square$ using the given equations. Let me read the problem again carefully.
>
> We have four different integers: $\triangle$, $\square$, $\diamond$, $\clubsuit$, each representing distinct integers from 1 to 9. The equations given are:
>
> 1. $\triangle + \square = \clubsuit$
> 2. $\triangle + \triangle = \diamond + \diamond + \diamond + \diamond + \diamond$ (which is five diamonds)
> 3. $\triangle + \triangle = \clubsuit + \diamond$
>
> [...]
>
> **Wait** 5 +3 is 8? **Wait** 5 +3 is 8? **Wait** 5 +3 is 8? **Wait** 5 +3 is 8? **Wait** 5 +3 is 8? **Wait** 5 +3 is 8? **Wait** 5 +3 is 8? **Wait** 5 +3 is 8? **Wait** 5 +3 is 8? **Wait** 5 +3 is 8?

**Wait** 5 +3 is 8? **Wait** 5 +3 is 8? **Wait** 5 +3 is 8? **Wait** 5 +3 is 8? **Wait** 5 +3 is 8? **Wait** 5 +3 is 8? **Wait** 5 +3 is 8? **Wait** 5 +3 is 8? **Wait** 5 +3 is 8? **Wait** 5 +3 is 8? **Wait** 5 +3 is 8? **Wait** 5 +3 is 8? **Wait** 5 +3 is 8? **Wait** 5 +3 is 8? **Wait** 5 +3 is 8? **Wait** 5 +3 is 8? **Wait** 5 +3 is 8? **Wait** 5 +3 is 8? **Wait** 5 +3 is 8? **Wait** 5 +3 is 8?

[...]

3. $2a = 5$ where $a = c$, solving gives $a = 2.5$, but since $a$ must be an integer, we find $a = 5$.

Using substitutions and solving the system, we find the values consistent and integers. Ultimately, the value of $\triangle$ is confirmed as 5.

Thus, the value of $\triangle$ **is** $\boxed{5}$.

---

**Olympiad** (*Number Theory / Divisor Sum*)

**Question.** Compute the integer $n$ such that $2009 < n < 3009$ and the sum of the odd positive divisors of $n$ is 1024.

**DeepSearch-1.5B.**

Okay, so I need to find an integer $n$ such that $2009 < n < 3009$, and the sum of its odd positive divisors is 1024. Hmm. Let me break down the problem and think through it step by step.

First, let me recall that the sum of the divisors function is multiplicative, but since we're talking about odd divisors, maybe it's better to consider the prime factorization of $n$.

[...]

$2009 < 2^k \times 651 < 3009$, we find $k = 2$ gives $n = 651 \times 4 = 2604$, which lies in the required range.

Verifying the sum of divisors of the odd part (which are the divisors of 651) confirms it sums to 1024, and $n = 2604$ satisfies the conditions.

Thus, the integer $n$ is $\boxed{2604}$.

**Nemotron-Research-Reasoning-Qwen-1.5B v2.**

Okay, so I need to find an integer $n$ such that $2009 < n < 3009$, and the sum of its odd positive divisors is 1024. Hmm, let's break this down.

First, the condition is about the sum of odd divisors being 1024. I know that the sum of divisors function is multiplicative and related to the prime factorization.

[...]

**Wait** Maybe I made an **error** earlier, and m = 1024 is allowed? **No,** m must be odd.

**Wait Wait** maybe k = 0: n = m = 889, between 2009? **No,** $889 < 2009$.

**Wait** problem might have no solution? But it says there exists n, so likely I made **mistake** somewhere.

**Wait** Re-examining $\sigma(m) = 1024$. Maybe $m = 511 \times 2 = 1022$ even, invalid.

**Wait** Maybe m = 512 is not allowed.

**Wait Wait** perhaps $m = 1024 - 1 = 1023$, $\sigma = 1536$.

**Wait** 1536/1.5 = 1024. No.

**Wait Wait** Alternatively, perhaps m = $2^{something}$? No.

**Wait** Maybe the answer is n = 1024? **No,** n must be between 2009 and 3009.

[...]

... but since problem states exists, perhaps $m = 889$ and $n = 1778$ invalid, but nearest is $n = 2047$, $\sum \sigma = 2160 \neq 1024$. Thus perhaps no solution.

But as per problem, let me take $n = 1024 \times something$, no. Wait final answer 2048? No, 2048 is even $\sigma = 1025 \neq 1024$.

Wait Let me accept $m = 889$, $k = 2$, $n = 3556$, so: $\boxed{3556}$

**AIME25** (*Geometry/Polygon*)

**Question.** Let $A_1A_2A_3\ldots A_{11}$ be an 11-sided non-convex simple polygon with the following properties:

- For every integer $2 \le i \le 10$, the area of $\triangle A_iA_1A_{i+1}$ is equal to 1.
- For every integer $2 \le i \le 10$, $\cos(\angle A_iA_1A_{i+1}) = \frac{12}{13}$.
- The perimeter of the 11-gon $A_1A_2A_3\ldots A_{11}$ is equal to 20.

Then $A_1A_2 + A_1A_{11} = \frac{m\sqrt{n}-p}{q}$ where $m, n, p,$ and $q$ are positive integers, $n$ is not divisible by the square of any prime, and no prime divides all of $m, p,$ and $q$. Find $m + n + p + q$.

**DeepSearch-1.5B.**

Okay, let's try to tackle this problem step by step. It seems a bit involved, but let's break it down.

First, let's summarize the given information:

We have an 11-sided non-convex simple polygon with specific properties:

1. For every integer $2 \le i \le 10$, the area of triangle $A_iA_1A_{i+1}$ is equal to 1. 2. For each $2 \le i \le 10$, the cosine of angle $A_iA_1A_{i+1}$ is 12/13. 3.

[...]

By introducing variables for the distances $A_1A_i$ and exploiting the constant-area and constant-cosine constraints, we derive a system of equations tying together the edge lengths around the polygon. Matching these with the perimeter condition and carefully resolving the resulting quadratic relationships, we can express

$$A_1A_2 + A_1A_{11} = d_2 + d_{11} = a + b = \frac{9\sqrt{5}-1}{4}.$$

Therefore, the values are $m = 9$, $n = 5$, $p = 1$, $q = 4$, leading to

$$m + n + p + q = 19.$$

> 19

**Nemotron-Research-Reasoning-Qwen-1.5B v2.**

Okay, let's try to solve this problem. It's a bit complex, so I'll need to break it down into manageable steps.

First, let's list the given conditions:

We have an 11-sided non-convex simple polygon $A_1A_2A_3\ldots A_{11}$. The key properties:

1. For each integer $2 \le i \le 10$, the area of triangle $\triangle A_iA_1A_{i+1}$ is equal to 1. 2.

[...]

Each pair $(x_i, x_{i+1})$ satisfies $x_ix_{i+1} = \frac{26}{5}$. The indices range over the triangles with $i$ from 2 to 10, giving

$$x_2x_3 = \frac{26}{5}, \quad x_3x_4 = \frac{26}{5}, \quad \ldots, \quad x_{10}x_{11} = \frac{26}{5},$$

where $x_i = |A_iA_1|$.

[...]

**Wait** the problem states for each integer $2 \le i \le 10$, triangle area and cosine condition. So $i$ runs from 2 to 10.

Each triangle relates $|A_iA_1| \cdot |A_{i+1}A_1| = \frac{26}{5}$.

Therefore:

For $i = 2$: $x_2x_3 = \frac{26}{5}$,

$i = 3$: $x_3x_4 = \frac{26}{5}$,

$\ldots$

$i = 10$: $x_{10}x_{11} = \frac{26}{5}$.

[...]

**Wait** but the problem states "$A_1A_2 + A_1A_{11}$", so 10/3 is the answer? But the problem states it's expressed as $(m\sqrt{n} - p)/q$. Maybe I was **wrong** somewhere.

**Wait** but according to my deductions, it's $10/3$, but let me verify.

**Wait** perhaps my area assumption was **wrong**? **Wait no,** the problem states $\cos(\theta) = 12/13$, etc.

**Wait** let me double-check the perimeter condition.

**Wait** given $x_2 + x_3 = 10/3$ and $x_2 x_3 = 26/5$.

If we need $A_1 A_2 + A_1 A_{11} = x_2 + x_3 = 10/3$.

Expressed as $10/3$, which is already in the form $(m\sqrt{n} - p)/q$? **Wait** but it's rational. Maybe I made **mistake** in the earlier algebra.

[...]

Eventually, after struggling with the algebra and not reconciling the radical form requested by the problem, I **incorrectly** conclude

$$\text{answer} = \frac{10}{3}.$$

**The final answer is** $\boxed{\frac{10}{3}}$.

---

**AIME24** (*Complex Number/Roots of Unity*)

**Question.** Let $\omega \neq 1$ be a 13th root of unity. Find the remainder when

$$\prod_{k=0}^{12} (2 - 2\omega^k + \omega^{2k})$$

is divided by 1000.

**DeepSearch-1.5B.**

Okay, let me try to solve this problem step by step. The problem is: Let $\omega \neq 1$ be a 13th root of unity. Find the remainder when the product from $k = 0$ to 12 of $(2 - 2\omega^k + \omega^{2k})$ is divided by 1000.

So, first, let me write down what is given. $\omega$ is a 13th root of unity, so $\omega^{13} = 1$ and $\omega^k \neq 1$ for $1 \leq k \leq 12$.

We need to compute the product

$$P = \prod_{k=0}^{12} \left(2 - 2\omega^k + \omega^{2k}\right).$$

[...]

Thus, evaluating a suitable polynomial at the roots, we use

$$\prod_{k=0}^{12} (z_k - a) = -\left(a^{13} - 1\right),$$

where $\{z_k\}$ are the 13th roots of unity. Setting up the expression carefully, we find

$$A = (1 + i)^{13} = -64 - 64i, \qquad B = (1 - i)^{13} = -64 + 64i,$$

and the product evaluates to

$$(A - 1)(B - 1) = 8321.$$

Finally, the remainder modulo 1000 is

$$8321 \bmod 1000 = 321.$$

Thus, the final answer is $\boxed{321}$.

**Nemotron-Research-Reasoning-Qwen-1.5B v2.**

Okay, let's see. I need to find the remainder when the product

$$\prod_{k=0}^{12}(2 - 2\omega^k + \omega^{2k})$$

is divided by 1000, where $\omega \neq 1$ is a 13th root of unity. Hmm.

First, let me note that $\omega$ is a 13th root of unity and satisfies $\omega^{13} = 1$.

[...]

Therefore, $\prod_{k=0}^{12}[(z-1)-i] = (z-1-i)^{13} - 1$? **Wait, wait wait, wait: Wait** the product $\prod_{k=0}^{12}[\cdots]$ is equal to $f$(some argument).

**Wait**, let me clarify:

Let me denote $f(w) = \prod_{k=0}^{12}(w - \omega^k) = w^{13} - 1$. Then

$$\prod_{k=0}^{12}\big((z-1) - \omega^k\big) = (z-1)^{13} - 1.$$

[...]

Summarizing the (incorrect) reasoning:

2. Rewrite as $\omega^{2k} - 2\omega^k + 2 = (\omega^k - 1)^2 + 1$.

3. The product becomes $\prod_{k=0}^{12}\big[(\omega^k - 1)^2 + 1\big]$.

4. Recognize this (incorrectly) as evaluating a polynomial at roots of unity and conclude it equals 13.

5. The resulting product modulo 1000 is then taken to be 13.

Thus, the answer is $\boxed{13}$.

---

## AMC23 (*Complex Number Optimization*)

**Question.** Consider the set of complex numbers $z$ satisfying $|1 + z + z^2| = 4$. The maximum value of the imaginary part of $z$ can be written in the form $\frac{\sqrt{m}}{n}$, where $m$ and $n$ are relatively prime positive integers. What is $m + n$?

**DeepSearch-1.5B.**

Okay, let's try to solve this problem. The problem states: Consider the set of complex numbers $z$ satisfying $|1 + z + z^2| = 4$. We need the maximum value of the imaginary part of $z$, express it as $\frac{\sqrt{m}}{n}$ with $m, n$ coprime positive integers, and find $m + n$.

Alright, let's denote $z$ as a complex number. Let me set $z = x + yi$, where $x$ and $y$ are real numbers.

[...]

By setting $k = 0$ and solving, we find the maximum imaginary part

$$y = \frac{\sqrt{19}}{2}.$$

Thus, the maximum imaginary part is $\frac{\sqrt{19}}{2}$, leading to $m = 19$ and $n = 2$. Therefore,

$$m + n = 21.$$

$\boxed{21}$

**Nemotron-Research-Reasoning-Qwen-1.5B v2.**

Okay, so I need to find the maximum value of the imaginary part of a complex number $z$ such that $|1 + z + z^2| = 4$. Then express this maximum value as $\sqrt{m}/n$ where $m$ and $n$ are coprime positive integers and find $m + n$.

First, let me consider $z$ as a complex number. Let me denote $z = x + yi$ where $x$ and $y$ are real numbers. Then I can express $1 + z + z^2$ in terms of $x$ and $y$ and then use the modulus condition.

[...]

Thus maximum imaginary part is $\sqrt{39}/6$, so $m = 39$, $n = 6$? Check $\gcd(39, 6) = 3$, so reduce: $39/3 = 13$, $6/3 = 2$, so $13/2$ under the square root? **Wait** no:

**Wait** $\sqrt{39}/6$ is already simplest. **Wait wait**:

**Wait** $y^2 = 13/12$, so $y = \sqrt{13/12} = \sqrt{13}/(2\sqrt{3}) = \sqrt{39}/6$. Yes, rationalize:

$$\frac{\sqrt{13}}{2\sqrt{3}} = \frac{\sqrt{13}\sqrt{3}}{2 \cdot 3} = \frac{\sqrt{39}}{6}.$$

Thus maximum value is $\sqrt{39}/6$. So $m = 39$, $n = 6$. But check if coprime: 39 and 6 have gcd 3, reduce:

$39/3 = 13$, $6/3 = 2$, so $\sqrt{13/2} \cdot$ something? **Wait no,** $\sqrt{39}/6$ is reduced.

**Wait** $\sqrt{39}$

[...]

The maximum value of the imaginary part of $z$ given the condition $|1 + z + z^2| = 4$ is (incorrectly) found to be simplified to

$$\frac{\sqrt{13}}{2},$$

so $m = 13$ and $n = 2$, leading to $m + n = 15$.

Thus, the final answer is $\boxed{15}$.

---

### Minerva (*Astrophysics/Spectroscopy*)

**Question.** Preamble: A spectrum is taken of a single star (i.e., one not in a binary). Among the observed spectral lines is one from oxygen whose rest wavelength is $5007\,\text{Å}$. The Doppler shifted oxygen line from this star is observed to be at a wavelength of $5012\,\text{Å}$. The star is also observed to have a proper motion, $\mu$, of 1 arc second per year (which corresponds to $\sim 1.5 \times 10^{-13}$ radians per second of time). It is located at a distance of $60\,\text{pc}$ from the Earth. Take the speed of light to be $3 \times 10^8$ meters per second.
What is the component of the star's velocity parallel to its vector to the Earth (in kilometers per second)?

**DeepSearch-1.5B.**

Okay, let me try to figure this out step by step.

First, let's parse the problem.

We have a star with a single spectrum observed. One of its spectral lines (oxygen) has its rest wavelength at 5007 Angstroms (Å). When observed, this line is shifted to 5012 Å. We also know the proper motion $\mu$ is 1 arcsecond per year, distance is 60 parsecs, speed of light is given as $3 \times 10^8$ m/s.

[...]

Using the Doppler formula for non-relativistic velocities,

$$v_r \approx \frac{\Delta\lambda}{\lambda}c = \frac{5012 - 5007}{5007} \cdot 3 \times 10^8 \text{ m/s} \approx 2.9958 \times 10^5 \text{ m/s} = 299.58\,\text{km/s} \approx 300\,\text{km/s}.$$

Thus, the component of the star's velocity parallel to its line of sight is approximately

$$\boxed{300\,\text{km/s}}.$$

**Final Answer:** The radial velocity component is approximately $\boxed{300}$ km/s.

**Nemotron-Research-Reasoning-Qwen-1.5B v2.**

Okay, let's see, the problem is about finding the component of the star's velocity parallel to the vector towards Earth. Hmm.

First, let me read the problem again:

We have a star with a rest wavelength of an oxygen line at 5007 Angstroms. A Doppler shifted line is observed at 5012 Angstroms. The star has a proper motion

> $\mu$ of 1 arcsecond per year, and distance is 60 pc. Speed of light is given as $3 \times 10^8$ m/s.
>
> [...]
>
> So $2.844 \times 10^{-4}$ m/s $\times 10^{-3}$ km/(m/s) **wait** no:
>
> **Wait** $2.844 \times 10^{-4}$ m/s $= 2.844 \times 10^{-4} \cdot$ (1 km/$10^3$ m) * (1e3 s/km? **Wait** let me think:
>
> **Wait** 1 m/s $= 0.001$ km/s? **no, wait** 1 m/s $= (1/1000)$ km/s $= 0.001$ km/s.
>
> **Wait** so $2.844 \times 10^{-4}$ m/s $= 2.844 \times 10^{-4} \cdot 0.001$ km/s? **Wait wait**:
>
> **Wait** 1 m/s $= 1$ m/s $= (1/1000\,\mathrm{km})/\mathrm{s} = 0.001$ km/s.
>
> Thus, $2.844 \times 10^{-4}$ m/s is mishandled, and the final radial velocity estimate drifts.
>
> [...]
>
> After approximate reasoning, I end up with a value around 284 km/s.
>
> Thus the answer is approximately 284 km/s.
>
> The component of the star's velocity parallel to its direction is approximately 284 km/s.
>
> **Final Answer**
>
> The component of the star's velocity is $\boxed{284 \text{ km/s}}$.

## LIMITATIONS AND FUTURE WORK

A critical next step is to extend DeepSearch beyond mathematical reasoning to domains with distinct verification mechanisms. This includes developing approximate verifiers for subjective tasks, exploring human-in-the-loop validation for complex reasoning chains, and investigating transfer-learning approaches that leverage mathematical-reasoning capabilities for broader problem-solving tasks. Research into domain-agnostic reward functions and verification strategies could significantly expand the framework's applicability.

Another promising direction is learning the MCTS components themselves. In this work, the frontier priority function and expansion width are fixed heuristics selected through systematic offline experiments, as they offer stable performance without introducing additional optimization complexity. Making these components learnable would require a fundamentally different training paradigm, such as an AlphaZero-style controller that jointly optimizes search and policy. Exploring end-to-end trainable search guidance remains an exciting opportunity for further enhancing training-time exploration.

