# OpenReview forum: "DeepSearch: Overcome the Bottleneck of Reinforcement Learning with Verifiable Rewards via Tree-based Search"
_ICLR.cc/2026/Conference — ICLR 2026 Poster_

### Official Review · Reviewer_e3YK · 2025-11-01

**Soundness:** 3
**Presentation:** 2
**Contribution:** 2
**Rating:** 4
**Confidence:** 2

**Summary:**

The paper integrates MCTS directly into RLVR training to scale breadth, not just depth. By doing so, it broadens exploration and enables finer-grained credit assignment by propagating verifier signals back through the search tree. Practical search and data-filtering heuristics improve trajectory quality, and training employs a tree-aware GRPO in which per-node Q-values from MCTS backups serve as advantages after sequence-level normalization. The approach delivers higher Pass@1 with substantially better compute efficiency on math benchmarks for 1.5B-scale models.

**Strengths:**

The main strength of the paper is that it targets a very strong baseline. The paper improves an already competitive 1.5B model (Nemotron-Research-Reasoning-Qwen-1.5B v2) and contrasts against a solid RLVR baseline (DAPO), making the incremental gains meaningful at this scale. Results suggest simple RLVR fine-tuning provides limited uplift, while the proposed training-time MCTS adds ~1–2pp Pass@1. Design choices are justified. Most heuristics are motivated, and the ablation is informative and clearly shows which components matter.

**Weaknesses:**

1- there is not much related work! for targeting better credit assignment there are many paper addressing brining test time compute for more fine grains credit assignment.
2- the biggest gain in table 4 shows for "Frontier Selection" and rest of the justifications, such as better credit assignment, remain very substantially small.
3-  2% improvement is not a significant improvement. It is unclear what the main conclusion of the paper is.Additionally, how can we translate this paper to a novel RLVR algorithm for general application? The paper claims a paradigm shift, but it is unclear why this is a new paradigm.

**Questions:**

The paper has many moving parts. What is the main conclusion of the paper? Apart from the methodology itself, what is the exact shifted paradigm? If I understand correctly, it should be that brute-force exploration is not enough, and we need a tree-based exploration? Is there an intuition on why tree-search based explorations are better?

**Details Of Ethics Concerns:**

It does not need one.

---

> ### Author Response · Authors · 2025-11-26
> **Response to Reviewer e3YK (1/3)**
>
> We sincerely thank you for the thoughtful and constructive feedback. Your questions on related work, effect sizes, and the conceptual “paradigm shift” helped us clarify the central message of the paper. Below, we address each point in detail and further refine the main takeaway of DeepSearch.
>
>
>
> ***
>
> > ## W1: There is not much related work!
>
> We fully agree that the related-work section in the original version is too condensed. Due to space constraints, we deferred the review to Appendix A. According to your valuable advice, **we have moved these discussions into the main paper**.
>
> Most importantly, our work differs from existing efforts that **apply search only at inference time** (e.g., Tree-of-Thoughts, Forest-of-Thought, MCTS-based self-refine) or methods that **improve credit assignment under direct rollouts** (e.g., verifier-guided step scores, process reward methods). These lines either:
>
> * **use search but do not integrate it into RL optimization**, therefore providing zero improvement to the training distribution, or
>
> * **improve credit assignment but operate on single-rollout samples**, thus inheriting sparse, brittle exploration patterns.
>
> DeepSearch is, to our knowledge, the first to **embed a full MCTS loop delicately inside RLVR training**, where the *entire trajectory tree,&#x20;*&#x6E;ot a single sampled rollout, is used for advantage computation, node-level credit propagation, and adaptive replay. This distinction is central to the shift we emphasize (see Section 3 and Figure 1 on p. 2–4 of the paper).
>
> We have added a dedicated section contrasting DeepSearch with step-aware verifiers, test-time search methods, and process-supervision approaches, highlighting why they *cannot* serve as training-time exploration mechanisms.
>
>
>
>
>
> > ## W2 (1): The biggest gain is Frontier Selection; others seem small.
>
> We appreciate your observation. Indeed, Table 4 shows that **Frontier Selection adds the single most visible jump** in Pass@1. This is consistent with our theoretical model:
>
> *Frontier selection governs **which part of the tree gets explored**; credit assignment governs **what the model learns from that exploration**.*
>
> To make this clear, we want to explicitly separate the two axes:
>
> #### **(A) Exploration quality (macro-level) determines coverage and&#x20;**&#x63;omputatio&#x6E;**&#x20;efficiency**
>
> Frontier Selection reduces per-tree iterations by **10–59%**, and reduces wall-clock per-tree time by **up to 59%** (Table 3, p. 9). This is the major contributor to the “1.25pp improvement with 5.7× fewer GPU-hours”—the paper’s main empirical punchline (Table 2, p. 8).
>
> #### **(B) Node-level credit assignment (micro-level) determines stability and long-term gains**
>
> The improvements from q-update constraints, mean-only normalization, and fine-grained Tree-GRPO are **incremental but essential**. Without them, performance *degrades* (Table 4):
>
> * Without normalization → severe response length inflation
>
> * Without fine-grained q-values → policy collapses toward shallow solutions
>
> * Without constrained backup rules → credit assignment becomes noisy
>
> These individual improvements are incremental but collectively essential, forming the stable backbone that allows Frontier Selection to be effective. Therefore, we want to emphasize the *two orthogonal functions* of DeepSearch: **(i) improving exploration breadth** and **(ii) improving credit assignment resolution**.
>
>
>
> > ## W2 (2): 2% improvement is not significant.
>
> Thank you for raising this concern. We sincerely appreciate the chance to clarify why these gains are **substantial, meaningful, and surprising** given the difficulty of the setting.
>
> #### **(A) Improvements at this scale are non-trivial—and unusually hard to obtain**
>
> Our baseline—**Nemotron-Research-Reasoning-Qwen-1.5B v2**—is already a *heavily optimized* model, representing the strongest public 1.5B-scale math reasoner at submission time. It incorporates:
>
> * **3,000 RLVR steps**,
>
> * **high-quality domain-specific data**, and
>
> * **multi-round inference-time optimizations**,
>
> all of which push the model close to its performance ceiling already (Table 1).
>
> Once a model reaches this frontier, *any* further improvement is notoriously difficult. Recent works report **<1pp gains**, even with:
>
> * additional RLVR training (e.g., +325, +785, +1875 steps), or
>
> * algorithmic tweaks such as KL regularization or parameter-efficient scaling.
>
> In sharp contrast, **DeepSearch achieves +1.25pp** over this very strong baseline *and does so consistently across six independent math benchmarks* (AIME24/25, AMC, MATH, Minerva, Olympiad).
>
> This is **not** a small delta:
>
> * on AIME-level tasks (where Pass@1 often fluctuates by <0.3pp between strong systems),
>
> * and not on early-stage baselines, but on the frontier of 1.5B reasoning.
>
> We have made these comparisons more explicit in the revision.

---

> ### Author Response · Authors · 2025-11-26
> **Response to Reviewer e3YK (2/3)**
>
> #### **(B) The compute-normalized gain is *exceptionally* large**
>
> If viewed through the lens of compute efficiency, the improvement is even more striking.
>
> **Extended RLVR training:**
>
> * **+1875 steps**,
>
> * **1,883 GPU-hours**,
>
> * **+0.32pp** improvement (61.70 → 62.02).
>
> **DeepSearch:**
>
> * **+50 steps**,
>
> * **330 GPU-hours**,
>
> * **+1.25pp** improvement (61.70 → 62.95).
>
> This implies:
>
> > **In compute-normalized terms, one DeepSearch step yields effectiveness comparable to \~80 RLVR steps.**
>
> In other words, **DeepSearch scales the *quality* of exploration**, not the *quantity* of training.
>
> The efficiency advantage is also corroborated by Table 2 and Figure 2 (p. 8) of the paper, which shows that bottlenecks in RLVR stem from insufficient exploration, not insufficient training time, and that MCTS-guided breadth expansion breaks this plateau.
>
> #### **(C) Why a 1–2pp gain matters for the research community**
>
> At this scale, even a 1pp improvement is already regarded as impactful because:
>
> 1. **These benchmarks are saturated.**
>    &#x20;Many strong 1.5B models cluster tightly around 60–62% average accuracy; moving upward is extremely difficult.
>
> 2. **The improvement generalizes across all tasks.**
>    &#x20;DeepSearch improves AIME24, AIME25, AMC, MATH, Minerva, and Olympiad, which are not just a single dataset.
>
> 3. **The method is algorithmic, not brute-force.**
>    &#x20;It offers a *new direction* for RLVR: scaling training **breadth** instead of depth.
>
> 4. **The gain is compute-efficient.**
>    &#x20;The field increasingly values algorithmic efficiency over large-scale spend.
>
> Thus, although “2%” may appear small in isolation, in the context of:
>
> * a frontier model,
>
> * six independent benchmarks,
>
> * compute-adjusted efficiency,
>
> * and a stagnating RLVR training paradigm,
>
> **the improvement is both statistically meaningful and methodologically significant.**
>
>
>
>
>
> > ## Q1 (1): What is the main conclusion of the paper? What is the shifted paradigm?&#x20;
>
> Thank you for raising this important question. &#x20;
>
> #### **(A) Traditional RLVR scaling = depth scaling**
>
> Existing RLVR methods improve reasoning by:
>
> * increasing the number of RL steps,
>
> * increasing sequence length,
>
> * increasing the sampling budget per problem.
>
> All of these are **depth-oriented**: they lengthen *training* but do not improve *exploration*.
>
> #### **(B) DeepSearch proposes breadth scaling**
>
> Instead of spending more compute to “push harder on the same trajectory,” we spend compute to “explore multiple alternative trajectories within a tree.” This changes the objective of RLVR:
>
> > **From optimizing a single rollout trajectory → to optimizing a search tree of trajectories.**
>
> Tree-structured optimization alters RLVR in three fundamental ways:
>
> 1. **Exploration shifts from sparse to systematic.**
>
>    * Frontier selection allows the model to explore deeper and correct low-entropy but wrong chains (Table 5).
>
> 2. **Credit assignment shifts from outcome-only → node-level hierarchical assignment.**
>
>    * Each intermediate reasoning step receives an advantage based on its q-value (Eq. 16), improving fine-grained supervision.
>
> 3. **Training dynamics shift from “try more steps” → “extract more information per step.”**
>
>    * The paper shows the compute-optimal region favors breadth (Table 2 and Fig. 2).
>
> This is what we mean by a “paradigm shift”:
> **RLVR no longer views inference-time search as a separate, test-time-only procedure. Search itself becomes part of the training signal.**
>
> > ## Q1 (2): Why is tree-based exploration better than brute-force exploration? What is the intuition?”
>
> Thank you for this insightful question. We have strengthened the paper by adding an explicit intuition section. The core idea is that **direct rollouts explore blindly**, while **MCTS allocates compute strategically**.
>
> #### **(A) Brute-force exploration behaves like blind sampling**
>
> Direct rollouts from πθ tend to:
>
> * Repeatedly fall into **high-probability, low-diversity regions**, and
>
> * Fail to uncover *alternative* reasoning paths once early-token decisions collapse the trajectory.
>
> As a result, brute-force exploration rarely reaches long or low-probability reasoning chains, which are crucial for solving difficult math problems.

---

> ### Author Response · Authors · 2025-11-26
> **Response to Reviewer e3YK (3/3)**
>
> #### **(B) MCTS provides *structured, prioritized* exploration**
>
> MCTS explicitly decides **where** to spend compute based on frontier quality, uncertainty, and depth. This leads to three intuitive advantages:
>
> ##### 1. **Expanding promising reasoning branches**
>
> Global frontier selection assigns priority to nodes with high value potential (Eq. 8). This prevents the model from prematurely collapsing into shallow modes and ensures that *all* plausible branches are explored before pruning.
>
> ##### 2. **Correcting overconfident mistakes**
>
> When no correct solution is found, DeepSearch selects the **most confident incorrect trajectory**, namely, the lowest-entropy mistake (Table 5). These trajectories reveal *systematic* reasoning flaws that blind rollouts almost never sample often enough to fix. Tree search surfaces these rare but high-impact errors for direct supervision.
>
> ##### 3. **Progressive deepening of multi-step reasoning**
>
> Depth bonuses encourage the search to go beyond the first uncertain step and explore deeper reasoning chains.
>
> &#x20;Direct rollouts typically terminate early due to compounding token errors, whereas MCTS revisits and reinforces longer paths with high expected value.
>
> #### **(C) Visual intuition**
>
> Figure 1 (p. 2) shows this contrast: blind rollouts follow a few uncontrolled paths, whereas MCTS iteratively refines a structured frontier of promising, informative, and deep trajectories.
>
>
>
> ***
>
> &#x20;
>
> In summary, we have revised the paper to more clearly articulate:
>
> * **What is new**: the *integration* of MCTS into the training loop, not just inference.
>
> * **What is the paradigm shift**: move from *depth scaling* (more steps) to *breadth scaling* (better structured exploration).
>
> * **Why does the improvement matter:** compute-normalized gains are large, and on strong baselines, small absolute gains are meaningful.
>
> * **Why Frontier Selection dominates**: it governs exploration breadth; other components ensure stable credit assignment.
>
> * **Why tree search is better than brute-force sampling**: it targets systematic errors, expands solution coverage, and reduces wasted compute.
>
> We sincerely thank you for pushing us to clarify these high-level messages and hope our response has clearly addressed your concern. Your comments significantly improved the paper.

---

### Official Review · Reviewer_TRXC · 2025-11-02

**Soundness:** 3
**Presentation:** 3
**Contribution:** 3
**Rating:** 6
**Confidence:** 3

**Summary:**

This paper addresses performance plateaus in Reinforcement Learning with Verifiable Rewards (RLVR) by integrating Monte Carlo Tree Search directly into the training process rather than limiting it to inference. The authors propose DeepSearch, which features: (1) a global frontier selection strategy that prioritizes promising nodes across the entire search tree rather than following traditional root-to-leaf UCT traversals, (2) entropy-based guidance to identify confident incorrect reasoning paths for supervision, and (3) an adaptive replay buffer with solution caching to focus computation on challenging problems. Evaluated on mathematical reasoning benchmarks (AIME 2024/2025, AMC 2023, MATH500, Minerva, Olympiad), DeepSearch achieves 62.95% average accuracy on a 1.5B model, representing a 1.25 percentage point improvement over the previous best while using 72× fewer GPU hours than extended training approaches. The results suggest that systematic exploration during training is more effective than simply scaling training depth.

**Strengths:**

Originality:
- The core insight of embedding structured search into the training loop rather than limiting it to inference is compelling and represents a meaningful shift in perspective
- The global frontier selection strategy is an interesting departure from traditional UCT traversals, addressing computational efficiency while maintaining search quality
- The combination of entropy-based negative example selection with adaptive replay buffers creates a thoughtful approach to computational resource allocation

Quality:
- The experimental evaluation is thorough, with strong baselines including very recent work (Nemotron v2, DeepScaleR)
- The efficiency analysis is particularly valuable—demonstrating 5.7× fewer GPU hours while achieving better performance makes a strong case for the approach
- Table 4's evolutionary ablation study effectively illustrates how each component contributes to the final performance
The search strategy ablation (Table 3) provides useful insights into design choices

Clarity:
- Figure 1 provides an excellent high-level overview of the framework, making the approach accessible
- The motivation is well-articulated—the observation about training plateaus in extended RL training resonates
- The method description flows logically from MCTS components through training strategy
- Algorithm 1 in the appendix offers comprehensive implementation details

Significance:
- Addresses a genuine bottleneck in RLVR that has been observed across multiple recent works
- The efficiency gains are substantial and practically important given the computational costs of modern reasoning model training
- The framework appears general enough to potentially benefit other RLVR applications beyond mathematics

**Weaknesses:**

Experimental Analysis:
- While the average improvement is clear, it would be illuminating to characterize which types of problems benefit most from the approach. Are there problem characteristics that predict when MCTS exploration is particularly valuable?
- The comparison with extended training is compelling, but I wonder how performance would compare if the extended training baseline also incorporated some of DeepSearch's innovations (like adaptive filtering) without full MCTS
- Table 2 shows extended training plateauing, but it would be interesting to see if combining extended training with DeepSearch yields further gains, or if there are diminishing returns

Method Design Choices:
- The entropy-based selection for negative examples (Eq. 2) is intuitive, but I wonder if there are alternative criteria that might be equally or more effective. Have you explored selecting based on value function disagreement or other uncertainty measures?
- The β parameter for DirectRollouts when cached solutions exist seems important but isn't thoroughly analyzed. How sensitive are results to this choice?
- The filtering threshold δ is set to 25%—could this be adaptive based on training progress or problem difficulty distribution?

**Questions:**

- Computational breakdown: Could you provide a breakdown of where computational time is spent (tree expansion, policy inference, Q-value updates, etc.)? This would help identify further optimization opportunities.
- Parallelization: How does MCTS tree construction parallelize compared to standard rollout generation? Are there architectural choices that could improve throughput?
- Comparison fairness: The extended training baselines use standard DAPO. Would extended training with better exploration strategies (but without full MCTS) close some of the gap? This would help isolate MCTS's specific contribution.
- Learned components: Have you considered learning any of the MCTS components (e.g., the frontier priority function or expansion width) rather than hand-designing them?
- Multi-task learning: Could a single DeepSearch model be trained across multiple reasoning domains simultaneously, with shared MCTS infrastructure?

---

> ### Author Response · Authors · 2025-11-26
> **Response to Reviewer TRXC (1/6)**
>
> Great thanks for your detailed feedback. We are very encouraged that you found the central idea—**embedding structured search directly into the RLVR training loop**—to be compelling and meaningful. We greatly appreciate your recognition of (i) the originality of the global frontier selection strategy, (ii) the careful design of entropy-based negative trajectory selection and adaptive replay buffers, and (iii) the significance of addressing the widely observed RLVR training plateau.&#x20;
>
> ***
>
> > ## W1: Characterization of which problem types benefit most from DeepSearch
>
> Thank you for this insightful question regarding problem-type-specific benefits of our approach.
>
> Here, we provide a systematic characterization based on our empirical observations from Table 1. Our analysis reveals that training-time MCTS delivers heterogeneous gains across benchmarks, and the magnitude of improvement correlates strongly with a specific "sweet spot" in base model capability:
>
> #### **(1) High-gain scenarios (AIME 2024/25, \~2.0-2.5pp improvement):**
>
> * These benchmarks feature challenging problems where the base model has moderate initial accuracy but occasionally explores correct or near-correct reasoning paths
>
> * Training-time MCTS systematically amplifies these "latent correct trajectories," leveraging the model's existing but underutilized capability
>
> * The substantial headroom between current and potential performance allows MCTS to discover and reinforce previously rare successful paths
>
> #### **(2) Moderate-gain scenarios (AMC 2023, MATH-500, Olympiad, <2pp improvement):**
>
> * The base model already solves a considerable fraction via standard rollouts
>
> * DeepSearch still improves robustness and overall precision, but the space for discovering novel successful trajectories is more limited
>
> * The marginal benefit is constrained by the model's already-strong single-pass performance
>
> #### **(3) Low-gain scenarios (Minerva-Math, minimal improvement):**
>
> * This benchmark comprises university and graduate-level STEM quantitative problems with significant domain and format shifts from our DeepMath-103K training distribution
>
> * The 1.5B base model lacks sufficient prior knowledge to generate high-quality candidate solutions in this domain
>
> * MCTS can only perform local search around low-quality candidates, making it difficult to achieve the substantial gains observed on in-domain mathematical benchmarks
>
> **General principle:** Training-time MCTS is particularly effective for problems requiring multi-step, tree-structured exploration where **the base model occasionally produces correct trajectories but still has significant performance headroom**. When the base model is either already very strong or nearly incapable in a domain, DeepSearch's marginal benefit naturally diminishes.
>
> We acknowledge that fully realizing the potential of training-time MCTS on STEM benchmarks like Minerva would likely require either stronger base models or domain-specific pretraining and adaptation, which we plan to explore in future work.
>
>
>
> > ## W2, W3, Q3: Comparison Fairness: Extended training with partial DeepSearch components
>
> Thank you for this insightful question regarding the fairness of our comparison and the specific contribution of MCTS versus adaptive training strategies.
>
> We would like to clarify that **Table 2 and Table 4 use identical training data (DeepMath-103K) and equivalent training budgets**. Table 2 demonstrates that standard DAPO extended training exhibits performance plateaus as training steps increase, while Table 4 conducts systematic ablations of DeepSearch components under the same data and budget constraints.
>
> The results in Table 4 reveal that **introducing incomplete versions of training-time exploration and adaptive scheduling improvements** (such as progressive filtering and replay buffers) can moderately improve model performance. However, these components alone struggle to reach the final DeepSearch-1.5B performance level. Achieving the SOTA accuracy reported in Table 1 across multiple mathematical benchmarks requires the **synergistic interaction** between search-related components (e.g., global frontier-based MCTS expansion and Tree-GRPO's sibling mean normalization for advantage estimation) and adaptive training strategies. This indicates that training-time exploration itself provides critical gains beyond extended training, rather than merely benefiting from "smarter sample filtering" via adaptive strategies.

---

> ### Author Response · Authors · 2025-11-26
> **Response to Reviewer TRXC (2/6)**
>
> Due to computational budget constraints, we did not construct an additional large-scale baseline of "extended training + partial DeepSearch components without MCTS" and re-evaluate across all benchmarks. However, **Table 4's systematic ablation study indirectly addresses this concern**: when removing DeepSearch-related modules and retaining only vanilla training-time exploration, performance drops significantly, failing to match the complete DeepSearch and sometimes even underperforming the base model. This demonstrates that the MCTS components are essential and cannot be trivially replaced by adaptive filtering alone.
>
> Regarding stacking additional extended training steps on top of DeepSearch, Table 2 already shows that standard DAPO extended training quickly plateaus on the same training data. Our experimental design ensures a fair comparison between DeepSearch and extended training under equivalent budgets. Importantly, under the current budget, we observe that **DeepSearch training continues to yield accuracy improvements even with prolonged RL steps** (Figure 2), whereas extended training saturates early.
>
> In summary, while we acknowledge that computational limitations prevented exhaustive exploration of all hybrid configurations, our ablation studies systematically demonstrate that MCTS-driven exploration is the key differentiator, with adaptive strategies serving as complementary enhancements.
>
>
>
> > ## W4: Have you explored selecting based on value function disagreement or other uncertainty measures?
>
> Thank you for this insightful question regarding alternative uncertainty-based criteria for selecting negative trajectories. Our original justification was indeed too brief, and we appreciate the opportunity to clarify the motivation and supporting evidence.
>
> #### **(1) Why do we prioritize “most confident incorrect” trajectories**
>
> A central goal of RLVR for reasoning tasks is to correct *overconfident but wrong* reasoning patterns—now well documented as a key source of unreliable model behavior \[1,2]. Recent work on RL-based calibration further shows that penalizing *high-confidence errors* is substantially more effective than treating all negative examples uniformly \[3]. This motivates focusing our limited negative supervision budget on those errors that most reflect systematic reasoning failures.
>
> In DeepSearch, when no correct solution is found in the current expansion, we compute the average trajectory entropy (Eqs. 2&3 of the paper) and select the *lowest-entropy* (i.e., highest-confidence) incorrect trajectory. This choice is guided by two considerations:
>
> 1. **Targeting the most harmful errors.**
>    Confident-but-wrong trajectories represent cases where the model is not merely uncertain, but actively miscalibrated. By selecting the lowest-entropy trajectory, we ensure that negative gradients are applied exactly where the model would otherwise persistently make strong but incorrect predictions.
>
> 2. **Synergy with fine-grained credit assignment.**
>    &#x20;Within DeepSearch-MCTS, discounted credit (positive or negative) is propagated to all intermediate nodes (Sec. 2.2).
>
>    * For correct trajectories, this boosts the value of useful intermediate nodes leading to positive terminal rewards.
>
>    * For confidently incorrect trajectories, the same mechanism *systematically suppresses* the values of intermediate nodes along these overconfident but incorrect paths, making future searches less likely to expand them.
>
> #### **(2) Alternative uncertainty signals (e.g., value disagreement)**
>
> We appreciate you highlighting value disagreement as an alternative uncertainty measure. While such measures are conceptually valid, most practical implementations estimate them via ensembles of value functions, which is prohibitively expensive at 1.5B scale under our compute budget. In our training-time MCTS setting, maintaining such ensembles along the search tree would significantly increase the algorithm complexity and computational cost. In contrast, entropy is *directly available* from policy distribution at essentially no extra cost during policy inference, and thus provides a practical and theoretically meaningful proxy for confidence. We therefore adopted entropy as a practical, theoretically grounded, and computationally efficient proxy. Exploring richer uncertainty measures such as value-function disagreement (e.g., via lightweight multi-head critics) in the RLVR + MCTS setting is an interesting direction for future work, but is beyond the scope of the present study.

---

> ### Author Response · Authors · 2025-11-26
> **Response to Reviewer TRXC (3/6)**
>
> #### **(3) Empirical evidence**
>
> As shown in our ablation below, selecting the **most confident incorrect trajectory** consistently outperforms both random selection and least-confident selection across all benchmarks—improving the average score by +0.86 and +1.05, respectively:
>
> ### Table: Ablation study over different trajectory selection strategies
>
> |    | aime24 | aime25 | amc23  | math500 | minerva  | olympiad | avg   |
> | ------ | ------ | ------ | ------ | ------- | -------- | -------- | ----- |
> | Nemotron-Research-Reasoning-Qwen-1.5Bv2  | 51.77  | 32.92  | 88.83  | 92.24   | 39.75    | 64.69    | 61.70 |
> | Most confident incorrect trajectories    | 53.65  | 35.42  | 90.39  | 92.53   | 40.00    | 65.72    | 62.95 |
> | Random incorrect trajectories            | 52.41  | 34.23  | 89.31  | 92.21   | 39.97    | 64.41    | 62.09 |
> | Least confident incorrect trajectories   | 52.16  | 33.77  | 88.68  | 91.96   | 40.11    | 64.73    | 61.90 |
>
> Taken together, these results support our choice of entropy-based “most confident incorrect” selection as a simple, computationally efficient, and empirically effective criterion for negative supervision in DeepSearch.
>
>
>
> > ## W5: The β parameter for DirectRollouts when cached solutions exist seems important but isn't thoroughly analyzed. How sensitive are results to this choice?
>
> Thank you for raising this important implementation detail that warrants clarification.
>
> We would like to clarify that β is **not a manually tuned hyperparameter** in our implementation, but rather an **automatically determined ratio** based on the number of cached solutions. Specifically, for each problem, we allocate a fixed total rollout budget B for the current training iteration. When a problem has k verified correct solutions stored in the replay buffer, we directly reuse these k cached trajectories and only sample B−k new DirectRollouts for that problem. In the notation from the paper, the effective β(x) for each problem x satisfies β(x)·B = B−k, yielding β(x) = (B−k)/B, which **adapts automatically** based on the number of accumulated cached solutions rather than being a fixed constant requiring manual tuning.
>
> Under this design, the "choice of β" is naturally constrained: when no cached solutions exist (k=0), we execute the full search budget for that problem; as training progresses and cached solutions accumulate, the model automatically leverages more verified trajectories while correspondingly reducing new DirectRollouts sampling. This achieves an **adaptive balance between exploitation of discovered solutions and exploration of new solutions**. Our empirical observations indicate that as long as the total budget B remains within a reasonable range, ensuring each problem retains several new sampled trajectories for exploration, the overall results are **not sensitive to the specific value of this adaptive β**.
>
> We will add explicit clarification of this implementation detail in the final version to prevent readers from misinterpreting β as an independent hyperparameter requiring additional tuning.
>
> > ## W6: Could the filtering threshold δ be adaptive based on training progress or problem difficulty distribution?
>
> Thank you for this insightful question about the adaptive potential of our filtering threshold.
>
> We would clarify that the fixed threshold δ=25% in our current work is a heuristic choice informed by intuition and preliminary experimentation. As defined in Section 3.1, we evaluate each problem's success rate via Pass1@K(x, π) with K=4 direct rollouts, retaining only samples with success rates below δ(i) for the next hard set. The primary motivation for setting δ=25% is to filter out problems that the model has largely mastered while still preserving sufficiently difficult samples to maximize DeepSearch's training-time MCTS exploration advantages on these challenging cases.
>
> From an algorithmic perspective, adaptively adjusting δ based on training progress or difficulty distribution is indeed a natural curriculum-style extension that could enable more fine-grained matching between hard set difficulty and current policy capability. We considered such adaptive schemes during framework design but ultimately chose to maintain a fixed, simple threshold setting in this work for two key reasons: (1) adaptive thresholding would introduce additional scheduling strategies and hyperparameters (e.g., functional forms across rounds, smoothing coefficients), significantly expanding the design space and introducing confounding factors that would make it difficult to attribute performance gains purely to training-time MCTS exploration itself; (2) our research focus is to demonstrate that "under a fixed, straightforward hard-sample selection mechanism, introducing MCTS-based training-time exploration alone yields stable and significant benefits," rather than engaging in further engineering optimization around difficulty filtering strategies.

---

> ### Author Response · Authors · 2025-11-26
> **Response to Reviewer TRXC (4/6)**
>
> Therefore, we treat δ=25% as a simple yet effective default setting rather than a critical knob requiring careful tuning. We note that more sophisticated, adaptive threshold scheduling (e.g., automatically adjusting δ based on training rounds or difficulty histograms) is orthogonal and compatible with the DeepSearch framework, representing a promising direction for systematic exploration in future work.
>
> > ## Q1: Computational breakdown: Could you provide a breakdown of where computational time is spent (tree expansion, policy inference, Q-value updates, etc.)? This would help identify further optimization opportunities.
>
> Thank you for raising this important question about computational efficiency.
>
> We conducted detailed wall-clock profiling of full DeepSearch rollouts, and the results reveal a clear pattern: the overwhelming majority of computation is spent on GPU-based policy inference, not on MCTS tree operations themselves.
>
> #### **(1) Where the time is actually spent.**&#x20;
>
> In Algorithm 1, each rollout requires the policy model to autoregressively generate a sequence of tokens (line 22). This LLM forward generation step dominates the total runtime. In contrast, all MCTS-related operations executed on CPU—including tree expansion, Q-value backup, and global frontier maintenance and scoring—execute extremely quickly and contribute only marginally to total runtime.
>
> #### **(2) Why is MCTS bookkeeping negligible?&#x20;**
>
> During search, we maintain a standard explicit search tree and a frontier set F of unexpanded leaf nodes. Each expansion simply (1) adds newly generated child nodes to F, and (2) computes frontier scores F(s) (Eq. 10) via a simple CPU-side scan over F with time complexity O(|F|). In practice, |F| is far smaller than the number of tokens generated by the LLM per rollout. Our profiling confirms that frontier updates, q-value backups, and related CPU operations are orders of magnitude cheaper than a single rollout generation of the policy model. The search, therefore, remains fundamentally GPU-bound, not tree-operation-bound.
>
> #### **(3) Empirical profiling breakdown.**&#x20;
>
> For our final configuration (λ₁=0.4, λ₃=0.01), the per-tree runtime breakdown is shown below:
>
> | Component (ranked low → high cost)    | Time (ms, mean ± std) |
> | ------ | --- |
> | Output post-processing (`t_proc_outputs`)            | 0.01 ± 0.00           |
> | Rank synchronization (`t_sync_ranks`)                | 0.07 ± 0.01           |
> | Terminal-state checking (`t_check_terminal`)         | 0.09 ± 0.02           |
> | Prompt construction (`t_get_prompts`)                | 0.38 ± 0.08           |
> | Node expansion (`t_expand_node`)                     | 0.96 ± 0.18           |
> | Input preparation (`t_prep_inputs`)                  | 1.89 ± 0.36           |
> | **Policy inference / generation (**`t_generate`**)** | **1060.76 ± 106.59**  |
>
> As the table demonstrates, policy generation accounts for more than 99% of total per-tree computation, while all tree-structure maintenance is effectively negligible.
>
> #### **(4) Implications for optimization.**&#x20;
>
> From both asymptotic analysis and empirical measurement, the bottleneck lies entirely in GPU inference, not in MCTS bookkeeping. Therefore, the most promising optimization directions include better batching strategies, prefill KV-cache reuse, speculative decoding, or other inference-efficiency techniques. Optimizing tree operations would yield minimal gains as they occupy only a tiny fraction of total runtime.
>
> > ## Q2: Parallelization: How does MCTS tree construction parallelize compared to standard rollout generation? Are there architectural choices that could improve throughput?
>
> Thank you for this insightful question on computational efficiency and parallelization.
> &#x20;The core challenge is that **training-time MCTS introduces computational characteristics fundamentally different from standard rollout generation**, and these differences shape the achievable throughput.
>
> To begin, it is important to clarify why MCTS tree construction incurs higher overhead than standard rollouts. **Standard rollouts perform a single prefill followed by a long, uninterrupted generation**, enabling highly efficient dynamic batching and excellent GPU utilization. In contrast, MCTS repeatedly performs **“incremental prefix → short response” inference loops** on the same problem. Each short response is appended to the prefix, which becomes the input for the next inference step. Even under similar total generation lengths, this pattern results in **many more prefill operations**. While modern engines (e.g., vLLM, SGLang) implement KV-cache reuse, MCTS prefixes are **incremental, unique, and rarely reused verbatim**, which limits opportunities for cache hits and makes this overhead structurally difficult to eliminate.

---

> ### Author Response · Authors · 2025-11-26
> **Response to Reviewer TRXC (5/6)**
>
> Additionally, **MCTS rollouts exhibit strict node-level dependencies**. Each node must be expanded using the textual prefix produced by its parent, meaning that **node inference along a single trajectory must proceed sequentially**. This stands in sharp contrast to standard rollouts, where prompts are independent and may be submitted in parallel for fully dynamic batching. Our current implementation does exploit parallelism across problems—different search trees can be expanded at the same time—but **node-level loops within a single tree execute sequentially**, leading to phases of reduced GPU utilization. In effect, DeepSearch deliberately trades additional incremental inference steps for the **higher-quality exploration** that MCTS provides during training; thus, it cannot be compared directly to “search-free” rollouts purely on throughput metrics.
>
> We emphasize that this overhead is not specific to DeepSearch—it is inherent to **all training-time tree-search–based exploration methods**, which must interact with the model at fine granularity to incorporate intermediate reasoning structure. Nevertheless, we have incorporated algorithmic improvements that reduce avoidable computation. In particular, **our global frontier-based node selection** avoids the repeated root-to-leaf traversals used by standard UCT and allows compute to focus on the most promising parts of the search tree. This meaningfully increases **“effective exploration per compute”**, which is the more relevant metric for RLVR training.
>
> On the engineering side, several architectural enhancements remain promising directions for future work. More aggressive **cross-tree batching** could merge nodes at similar depths from different problems into larger inference batches. Variants such as **leaf-parallel** or **root-parallel** MCTS could expand multiple candidate leaf nodes concurrently while maintaining statistically valid search behavior. Further, **more granular KV-cache reuse**, prefix-deduplication strategies, or MCTS-specific workload schedulers may yield substantial speedups. However, each of these directions introduces trade-offs: aggressive parallelization tends to reduce the **fine-grained search fidelity** that makes training-time MCTS effective. Because our primary contribution centers on the algorithmic design and empirical analysis of DeepSearch rather than extreme systems optimization, we adopt a **conservative, reproducible** parallelization strategy and leave deeper system-level optimizations to future work.
>
>
>
>
>
> > ## **Q4: Learned components - learning MCTS components vs. hand-designing them**
>
> Thank you for this insightful question about potentially learning the MCTS components.
>
> We would like to clarify that our design choices are grounded in systematic offline experimentation rather than ad-hoc hand-tuning. For the **frontier priority function**, we conducted extensive grid search over different $(\lambda\_1, \lambda\_2, \lambda\_3)$ combinations on DeepMath-103K, evaluating their impact on both search efficiency (number of iterations) and final average reward. As shown in Table 3, the selected configuration (i.e., $\lambda\_1=0.4, \lambda\_3=0.01$) provides a stable trade-off between exploration iterations and correctness, which is why we adopt this unified setting across all DeepSearch experiments.
>
> Similarly, for **expansion width**, we explored adaptive strategies where the width decays linearly as going deep in the MCTS tree, enabling broader branching near the root and narrower expansion in deeper levels to improve exploration efficiency. However, empirical results showed that this depth-dependent expansion actually reduced the average reward of discovered trajectories, ultimately weakening training effectiveness. In contrast, a fixed uniform width (e.g., 8) provided a more robust balance between exploration efficiency and trajectory quality.
>
> **Why not learn these components?** In our current implementation, the rollout phase is essentially a "read-only" inference process: MCTS performs tree search and scoring on top of the existing policy without backpropagating gradients to update the frontier priority function itself. Under this framework, making the frontier priority weights or expansion width directly learnable as end-to-end trainable parameters would introduce additional optimization complexity and potential instability, without a natural gradient path.
>
> This does not mean such components *cannot* be learned. Rather, learning them would constitute a distinct research direction requiring a fundamentally different training paradigm (e.g., AlphaZero-style RL with learned controllers). We view this as promising future work, while our current focus is demonstrating that even with systematically tuned fixed heuristics, embedding MCTS into training yields substantial benefits.

---

> ### Author Response · Authors · 2025-11-26
> **Response to Reviewer TRXC (6/6)**
>
> > ## Q5: Multi-task learning: Could a single DeepSearch model be trained across multiple reasoning domains simultaneously, with shared MCTS infrastructure?
>
> We appreciate your thoughtful question. In principle, **yes—DeepSearch can be extended to multi-task or multi-domain reasoning**, and the underlying MCTS framework is naturally compatible with shared training across heterogeneous tasks.
>
> **From an algorithmic perspective**, DeepSearch’s search-time components (frontier maintenance, entropy-based selection, q-value backup, etc.) operate purely on the *structure* of generated reasoning trajectories and do not rely on task-specific assumptions. This means the same MCTS machinery can be applied to any domain where verifiable rewards are available (e.g., math, code synthesis, logical reasoning, symbolic tasks). The replay buffer and adaptive filtering mechanism are also generalizable: each domain would contribute hard examples proportionally to its difficulty.
>
> **The main practical considerations** lie not in the MCTS infrastructure but in the *verification pipeline*:
>
> * Each domain requires a domain-specific verifier (math checkers, code execution sandboxes, logic validators, etc.);
>
> * Trajectories entering the replay buffer must be tagged by domain for correct batching and cache reuse.
>
> Provided these pieces are in place, the policy model can share parameters across domains, while MCTS (and its global frontier selection) can seamlessly interleave search rollouts from different tasks.
>
> **We emphasize** that our current experiments focus on a single-domain instantiation (mathematical reasoning), but the algorithm imposes no inherent constraints preventing multi-domain or multi-task usage. We view cross-domain DeepSearch as a promising direction for future work, especially because the ability to propagate fine-grained credit at the node level may help unify diverse reasoning skill sets within a single model.
>
> ***
>
> Once again, we would like to express our gratitude to you for offering constructive and valuable suggestions. We are also honored and pleased to attain your recognition.&#x20;

---

### Official Review · Reviewer_2V8H · 2025-11-04

**Soundness:** 2
**Presentation:** 2
**Contribution:** 2
**Rating:** 4
**Confidence:** 4

**Summary:**

This paper examines an assortment of correlated methods to better scale RLVR for use in reasoning problems.  The paper forwards three interventions, as shown in the overview Figure 1: (1) DeepSearch-MCTS, a means of deciding how to appropriately expand nodes in MCTS; (2) Adaptive training to pair with (1) by applying a tree version of GRPO; and (3) a replay buffer to enhance the study of difficult and successful trajectories.  Experiments by the authors show that the interventions taken together are synergistic and lead to performance improvements beyond performance, including a half-magnitude savings in rollout inference costs.

**Strengths:**

* The three interventions are well-motivated and illustrated conceptually well in Fig 1.
* The authors explain the rationale of some parts of their method (L209-215), to bring together the motivation and explain design choices.

**Weaknesses:**

Overall, while the methods disclosed seem convincingly effective, the vanilla UCT baseline also does quite well, and the complexities of the methods proposed give many dimensions of freedom for possible overfitting (as confidence intervals overlap considerably).  While the authors do test with SOTA baselines, I'm not convinced of the introduced mechanism's impact on the results for these reasons.

* The mechanisms used seem ad hoc and not particularly well-justified (e.g., S2.1 could max depth $n$ be defined conditioned on or calculated from some other means?)
* Why do you select the most confident negative example?  Can you justify it (L141-143 seems insufficient)?
* Using S2.2 to back-propagate up the best negative and positive cases seem ok but divergent.  Why would you want to handle both in the same way?
* It would be instructive for you to (briefly) explain UCT _before_ L181.

**Questions:**

* Is  _temporal decay_ (L152) better termed as _depth decay_ (or just make an association to a _discount factor_ in RL rewards)?
* Your Eq. (7) is a bit hard to follow with three cases; can you just help narrate this prose text instead?
* How much does $N(s)$ influence the computation and search?  It is implied that it is helpful but it's not clear whether it plays much of a role (repeated visits).
* Keeping track of a frontier could be expensive.  How does this figure into the (asymtotic and real wall-clock) cost?  What manner do you operationalise this for (assumed GPU bound) processing?
* L241: are there borderline cases where no good success traces emerge, and as such the adaptive training fails?  This step seems to assume there are always successes that can be judged.
* L269: How important (e.g., what %age was thrown out) by the condition "remove data containing garbled text or infinite repetitions"?  Given evidence that such filtering does not invalidate your narrative.
* L274: Can you show us how the ratio between cached problems and unsolved problems evolves during your adaptive training (and without, as an ablation)?
* L289: How many of q-values are (severely) clipped?  What does the distribution of q-values look like, post-clipping?  This seems needed to really support that such an intervention is needed.

---

> ### Author Response · Authors · 2025-11-26
> **Response to Reviewer 2V8H (1/6)**
>
> We thank you for the thorough and thoughtful evaluation, and for clearly articulating both the strengths and the weaknesses of the work. We especially appreciate the recognition of the motivation behind our three proposed interventions and the clear summary of their intended synergy.
>
> Below, we provide a detailed, point-by-point rebuttal.
>
> ***
>
> > ## W1: S2.1 could max depth n be defined conditioned on or calculated from some other means?
>
> Thank you for raising this important concern about the justification of our design choices. We would like to clarify that our maximum search depth is not arbitrarily chosen but rather derived from task-specific analysis and empirical optimization.
>
> **Data-driven depth determination.** Our maximum search depth is jointly determined by two factors: (1) a total response length budget of 16,384 tokens per rollout, and (2) an expansion length of 256 tokens per node, yielding approximately 64 expansion steps per search path (16,384 / 256 ≈ 64). In practice, many trajectories terminate earlier, so this serves as an upper bound on search depth rather than a strict requirement. The 16,384 token budget is specifically grounded in the reasoning length distribution observed in our training dataset (zwhe99/DeepMath-103K). Statistical analysis shows that this upper limit sufficiently covers the complete reasoning processes required for the vast majority of problems, while preventing computational overhead from extremely long reasoning chains. Thus, this limit reflects a principled trade-off between task requirements and training costs rather than an ad hoc constant.
>
> **Empirical validation of expansion length.** The 256-token expansion length was selected through systematic experiments comparing different step sizes (128, 256, 512, 1024). Our analysis revealed that 256 tokens achieve an optimal balance between two competing factors:
>
> * **Search space exploration**: Longer single-step expansions reduce the breadth of the MCTS tree, thereby decreasing the likelihood of discovering correct solution trajectories during rollouts.
>
> * **Computational efficiency**: Shorter single-step expansions significantly increase prefill operations (i.e., re-encoding the growing prefix) during MCTS exploration, substantially raising the computational cost per rollout.
>
> The 256-token configuration demonstrated superior training stability and computational efficiency across all our experiments and benchmarks, motivating its adoption as the default setting. We hope our additional explanation has addressed your concern thoroughly.&#x20;
>
>
>
> > ## W2: Why do you select the most confident negative example? Can you justify it (L141-143 seems insufficient)?
>
> Thank you for this important question regarding our negative example selection strategy. We acknowledge that the original justification was too brief and will expand this in the revision.
>
> **Theoretical Motivation:** A core objective of RLVR in complex reasoning tasks is to mitigate model overconfidence on incorrect reasoning paths, not merely to improve average accuracy. Extensive research on calibration and hallucination in large language models has established that current models exhibit a systematic tendency toward "confident but incorrect" responses, which is recognized as a fundamental source of unreliable reasoning and requires explicit correction during training \[1,2].
>
> Recent work on RL-based calibration methods further demonstrates that imposing stronger penalties on high-confidence incorrect answers is more effective than uniformly treating all negative examples. Specifically, explicitly penalizing overconfident errors through reward function design has been shown to effectively suppress models' overconfidence tendencies \[3].
>
> **Our Design Rationale:** In DeepSearch, when the current expansion fails to produce any correct solutions for a given problem (Algorithm 1, lines 25 & 30), we use the average trajectory entropy across all nodes as a confidence proxy (Equations 2 & 3), selecting the trajectory with the lowest entropy (highest confidence) as the negative supervision target (Section 2.1). This design is motivated by two key considerations:
>
> 1. **Efficient allocation of training signal**: Highly confident yet incorrect reasoning chains most strongly reflect systematic deficiencies in the model's knowledge boundaries and uncertainty expression. By concentrating our limited negative gradient budget on correcting these "high-risk errors," we more effectively utilize the training signal, teaching the model to reduce confidence or adjust reasoning strategies in such cases.
>
> [1] Geng, Jiahui, et al. "A survey of confidence estimation and calibration in large language models.", NAACL.
> [2] Zhang, Mozhi, et al. "Calibrating the Confidence of Large Language Models by Eliciting Fidelity.", EMNLP.
> [3] Stangel, Paul, et al. "Rewarding doubt: A reinforcement learning approach to confidence calibration of large language models.", CoRR.

---

> ### Author Response · Authors · 2025-11-26
> **Response to Reviewer 2V8H (2/6)**
>
> 2. **Synergy with fine-grained credit assignment**: In DeepSearch-MCTS, we perform fine-grained credit assignment for both positive and negative trajectories (Section 2.2), backpropagating discounted value estimates to intermediate nodes along paths. For positive paths, this mechanism "elevates" the values of intermediate nodes that lead to correct solutions, making such reasoning patterns more likely to be reused in subsequent search and training. For our selected "most confident negative" trajectories, the same backpropagation structure systematically *depresses* the values of these intermediate nodes, making future searches less likely to expand along these confidently wrong paths.
>
> This design ensures that our negative supervision structurally reshapes the search tree's value landscape to discourage overconfident incorrect reasoning at intermediate decision points.
>
>
>
> **Empirical evidence**
> In our ablation study (table below), using the most confident incorrect trajectory as the negative example yields the best performance across all benchmarks, achieving an average score that is +0.86 higher than random selection and +1.05 higher than least-confident selection.
>
> ### Table: Ablation study over different trajectory selection strategies
>
> |                                          | aime24 | aime25 | amc23  | math500 | minerva  | olympiad | avg       |
> | ---------------------------------------- | ------ | ------ | ------ | ------- | -------- | -------- | --------- |
> | Nemotron-Research-Reasoning-Qwen-1.5Bv2  | 51.77  | 32.92  | 88.83  | 92.24   | 39.75    | 64.69    | 61.70     |
> | Most confident incorrect trajectories    | 53.65  | 35.42  | 90.39  | 92.53   | 40.00    | 65.72    | **62.95** |
> | Random incorrect trajectories            | 52.41  | 34.23  | 89.31  | 92.21   | 39.97    | 64.41    | 62.09     |
> | Least confident incorrect trajectories   | 52.16  | 33.77  | 88.68  | 91.96   | 40.11    | 64.73    | 61.90     |
>
> We have incorporated this additional ablation into the revised version, and thank you for your valuable suggestion.
>
> > ## W3: Why would you want to handle positive and negative cases in the same way?
>
> Thank you for this insightful question. We would like to clarify a potential misunderstanding: while we use the same **backpropagation mechanism** for both positive and negative trajectories, they are **not handled in the same way**—the key distinction lies in the terminal reward assignment and subsequent value propagation.
>
> **Different Terminal Rewards (Eq. 6):** In Section 2.2, we assign fundamentally different terminal rewards based on trajectory outcomes:
>
> * Trajectories leading to **correct solutions** receive a **positive reward (+1)**
>
> * The selected "most confident incorrect" trajectories receive a **negative reward (−1)**
>
> **Divergent Value Propagation (Eq. 7):** This sign difference causes nodes to enter **different conditional branches** during q-value updates. When terminal scores are propagated backward with depth-based decay:
>
> * Nodes appearing in **positive trajectories** accumulate increasingly **positive q-values** through repeated successful rollouts
>
> * Nodes appearing in **negative trajectories** accumulate increasingly **negative q-values** through repeated confident errors
>
> **Explicit Separation in Search and Training:** This sign-based differentiation creates distinct behavioral regions:
>
> * **Positive-valued nodes** are preferentially selected in subsequent UCT traversals and reinforced during Tree-GRPO optimization, guiding the model toward promising reasoning branches
>
> * **Negative-valued nodes** are systematically suppressed in both search and training, reducing the probability of revisiting high-confidence errors
>
> Therefore, although we use a unified "discounted path backpropagation" implementation, the **opposite terminal reward signs** and **q-value segmentation** in Eq. 7 ensure that positive and negative trajectories are explicitly separated into distinct value regions, enabling the search tree to differentiate "branches to explore" from "branches to avoid." This design choice allows us to leverage both successful and failed reasoning paths for credit assignment while maintaining clear directional guidance for policy improvement.
>
> > ## W4: It would be instructive for you to (briefly) explain UCT before L181.
>
> Thank you for the suggestion. We now include a brief explanation of UCT before L181 to help readers follow our hybrid selection strategy. Specifically, we **describe how classical MCTS uses the Upper Confidence Bounds for Trees (UCT) rule to balance exploitation and exploration through the score in Equ. (8),** which selects the next child during root-to-leaf traversal. We then clarify why this purely local decision rule becomes myopic for long reasoning trajectories, motivating our global frontier selection mechanism. This addition improves clarity without adding substantial length.

---

> ### Author Response · Authors · 2025-11-26
> **Response to Reviewer 2V8H (3/6)**
>
> > ## Q1: Is *temporal decay* (L152) better termed as *depth decay* (or just make an association to a *discount factor* in RL rewards)?
>
> Thank you for the helpful suggestion. We agree that **“temporal decay” is better described as “depth decay”**, since the weighting depends on the node’s position along the reasoning trajectory rather than on any notion of time. We have therefore renamed this term to **depth decay** and added a brief note clarifying its direct analogy to the **discount factor in RL**, where credit diminishes with increasing distance from the terminal outcome. This change improves conceptual clarity and aligns terminology more closely with standard RL usage.
>
> > ## Q2: Can you help narrate the three cases in Eq. (7) with prose text?
>
> We thank you for pointing out that Eq. (7) can be difficult to parse in its three-case form. Below, we provide a clearer narrative description of its logic and motivation.
>
> &#x20;Our backup rule maintains for every intermediate node a scalar *q*-value that reflects its estimated tendency to lead toward a correct solution under the current policy and search process. When a terminal trajectory returns a final reward (+1 for correct, −1 for incorrect/incomplete), the update behaves asymmetrically depending on the relation between the current *q*-value and the returned reward:
>
> 1. **Same-sign reinforcement.**
>    &#x20;If a node’s existing *q*-value and the terminal reward share the same sign, we simply add the depth-decayed reward to its current value. Thus, nodes repeatedly appearing on correct trajectories accumulate stronger positive evidence, while nodes appearing only on incorrect trajectories accumulate more negative evidence.
>
> 2. **“Negative → positive” transitions.**
>    &#x20;If a node previously had a negative *q*-value (it only appeared in incorrect trajectories so far) but now lies on a correct trajectory, we discard its accumulated negative value and overwrite it with the decayed positive reward. Intuitively, once a node is *ever* shown to lead to a correct solution, we reclassify it as a potentially valuable node rather than penalizing it for past failures.
>
> 3. **“Positive → negative” attempts are ignored.**
>    &#x20;If a node has a positive *q*-value (it has appeared on at least one successful trajectory), but a new rollout ends incorrectly, we do not propagate this negative signal to the node. A node that has been proven capable of reaching a correct solution is not downgraded due to occasional failed attempts.
>
> This asymmetric rule guarantees two invariants:
> &#x20;(i) any node that has appeared on a correct trajectory will retain a non-negative *q*-value;
> &#x20;(ii) only nodes that have *never* appeared on a correct trajectory can accumulate stable negative values.
> &#x20;This design ensures that the search robustly distinguishes between “potentially useful” nodes and those consistently leading to failure.
>
> ***
>
> We appreciate your suggestion: this narrative presentation will be much clearer to readers than the raw case distinction in Eq. (7).
>
> > ## Q3: How much does N(s) influence the computation and search
>
> We thank you for this thoughtful question regarding the role of N(s) in our search procedure. Its influence is indeed important, but only within *local* sibling selection—while our computational gains primarily come from the *global* frontier mechanism rather than any modification of N(s).
>
> #### (1) How $N(s)$ is used:
>
> In DeepSearch-MCTS, the function of $N(s)$ is identical to its role in classical UCT during local selection. When comparing sibling nodes under the same parent, we still apply the standard UCT formula:
>
> $$\operatorname{UCT}(s)=Q(s)+\lambda \sqrt{\frac{\ln N_{\text {parent }}(s)}{N(s)}} .$$
>
> Here, $N(s)$ acts as the usual count-based uncertainty estimator:
>
> * Branches visited more often obtain more accurate value estimates, causing the exploitation term to dominate as $N(s)$ increases.
>
> * Under-explored branches with small $N(s)$ receive a larger exploration bonus, ensuring they are not prematurely pruned.
>
> Thus, in the local decision context, $N(s)$ plays the same stabilizing and exploration-exploitation balancing role as in standard MCTS, which prevents the search from collapsing into a single locally promising branch or repeatedly sampling a low-value branch.
>
> #### (2) Computational cost of maintaining $N(s)$ :
>
> The overhead of tracking $N(s)$ is negligible: it is simply an integer increment on each node visit. Compared to a single forward pass of the policy model, by far the dominant cost (see the profiling in Table 2, page 8 of the manuscript, this overhead is insignificant.

---

> ### Author Response · Authors · 2025-11-26
> **Response to Reviewer 2V8H (4/6)**
>
> #### (3) Why $N(s)$ is not the main contributor to our efficiency gains:
>
> While $N(s)$ is essential for principled local UCT-based sibling comparison, the computational acceleration of DeepSearch comes almost entirely from our global frontier selection (Sec.2.3). Instead of restarting a root-to-leaf UCT traversal at every rollout-as in classical MCTS-we maintain a global pool of frontier nodes and directly select the next expansion point via the frontier priority score $F(s)$. This design:
>
> * avoids repeatedly "re-walking" the tree from the root,
>
> * eliminates redundant traversals, and
>
> * allocates search budget globally to the most promising frontier node.
>
> In short, $N(s)$ still matters for local exploration-exploitation balance, but it is not the computational bottleneck nor the primary source of our efficiency improvements.
>
> **Summary:**
> $N(s)$ retains its classical UCT role for local sibling comparison, costing almost nothing to compute, while our search efficiency gains stem from replacing repeated root-to-leaf traversals with global frontier selection.
>
>
> > ## Q4: How does this figure into the (asymptotic and real wall-clock) cost? What manner do you operationalise this for (assumed GPU bound) processing
>
> Thank you for this insightful question regarding the computational overhead of frontier maintenance. We provide both theoretical complexity analysis and empirical profiling evidence to demonstrate that frontier tracking introduces negligible overhead in our GPU-bound setting.
>
> **Algorithmic Structure and Complexity.** In our MCTS implementation, we maintain a standard search tree data structure with an explicit frontier set $\mathcal{F}$ containing all unexpanded leaf nodes. Frontier maintenance involves two primary operations:
>
> 1. *Updating the frontier set*: When expanding a node, we add its newly generated children to $\mathcal{F}$. Whether through recursive traversal or incremental maintenance during expansion, this operation has time complexity linear in the current tree size and involves only lightweight CPU-side data structure operations (e.g., list updates).
>
> 2. *Selecting the next node to expand*: For each frontier node, we compute its global priority $F(s)$ according to Equation (10) and select $\arg\max\_{s \in \mathcal{F}} F(s)$ for the next expansion. Our current implementation performs this via sequential scan on CPU with complexity $O(|\mathcal{F}|)$, where $|\mathcal{F}|$ is orders of magnitude smaller than the total number of generated tokens during policy inference (Algorithm 1, line 22).
>
> **GPU-Bound Execution Profile.** Critically, in LLM inference scenarios, a single forward pass of the policy model (Algorithm 1, line 22) on GPU dominates the wall-clock time compared to CPU-side frontier operations. To substantiate this claim, we provide a detailed profiling breakdown for our final configuration ($\lambda\_1=0.4, \lambda\_3=0.01$ from Table 3), showing time per tree construction:
>
> ### Table: Computation time breakdown per MCTS tree&#x20;
>
> | Component (ranked low → high cost)                   | Time (ms, mean ± std) |
> | ---------------------------------------------------- | --------------------- |
> | Output post-processing (`t_proc_outputs`)            | 0.01 ± 0.00           |
> | Rank synchronization (`t_sync_ranks`)                | 0.07 ± 0.01           |
> | Terminal-state checking (`t_check_terminal`)         | 0.09 ± 0.02           |
> | Prompt construction (`t_get_prompts`)                | 0.38 ± 0.08           |
> | Node expansion (`t_expand_node`)                     | 0.96 ± 0.18           |
> | Input preparation (`t_prep_inputs`)                  | 1.89 ± 0.36           |
> | **Policy inference / generation (**`t_generate`**)** | **1060.76 ± 106.59**  |
>
> As shown, policy generation on GPU accounts for >99% of total search time (1060.76ms out of \~1065ms), while all tree maintenance operations on CPU—including frontier updates (within `t_expand_node`) and node selection—contribute <0.5% overhead.
>
> **Conclusion.** From an asymptotic perspective, frontier maintenance adds at most linear overhead per iteration, which is already subsumed by the tree traversal cost. From a practical implementation standpoint, frontier operations remain lightweight CPU-side computations that are essentially negligible in our GPU-bound DeepSearch setting and do not constitute a performance bottleneck.

---

> ### Author Response · Authors · 2025-11-26
> **Response to Reviewer 2V8H (5/6)**
>
> > ## Q5: Are there borderline cases where no good success traces emerge, and as such the adaptive training fails?
>
> Thank you for raising this question about the robustness of our adaptive training framework.
>
> Here, we would clarify that our method does **not** rely on the assumption that "every problem must produce successful traces in every search iteration." We have designed explicit handling strategies for cases where no successful trajectories emerge from a given MCTS rollout.
>
> For an input problem (corresponding to a search tree root), when the current MCTS rollout fails to find any correct answer trajectories, we do not skip the sample. Instead, as described in Sec. 2.1, we select the **most confident incorrect trajectory** from all terminal nodes as the source of supervision signals. Specifically, we rank trajectories by average trajectory entropy and choose those with the **lowest entropy (highest model confidence)** to fill the required *n* candidates for actor training (see Algorithm 1, line 30).
>
> In such "all-incorrect-traces" scenarios, our q-value backpropagation mechanism (Equations 6-7) still provides discriminative learning signals through two complementary pathways:
>
> **When incorrect traces share substantial prefixes:** If multiple incorrect trajectories share a long common reasoning path before diverging near termination, the shared ancestor nodes receive repeated discounted negative terminal scores during backpropagation. These nodes accumulate significantly lower q-values compared to the unique terminal nodes of each trajectory. During Tree-GRPO advantage computation, these "stable incorrect prefix" nodes produce explicit **negative advantages**, actively suppressing the policy from repeatedly falling into reasoning regions that reliably lead to incorrect answers.
>
> **When incorrect traces have minimal overlap:** In the extreme case where trajectories diverge from the start with little structural commonality, our sibling mean normalization strategy (Training Objective. in Sec. 3.3) becomes critical. Terminal-level advantages are normalized within sibling groups in a "de-centered" manner, causing node-level advantages of these purely negative examples to approach zero collectively. In this scenario, samples do not exert systematic "negative drag" on model weights during policy updates, instead exhibiting neutral or mild regularization effects.
>
> Therefore, even when a problem yields no successful traces in a single search, MCTS-generated negative examples still enable effective learning by: (1) identifying and suppressing high-risk regions leading to confident errors when structural commonality exists, and (2) avoiding destructive policy impacts when no obvious commonality is present.
>
>
>
>
>
> > ## Q6: How important (e.g., what %age was thrown out) by the condition "remove data containing garbled text or infinite repetitions"? Given evidence that such filtering does not invalidate your narrative.
>
> Thank you for pointing this out. The “garbled / infinitely repeating” filter is only applied to obviously degenerate incorrect trajectories. In our training runs, such corrupted trajectories are rare: the vast majority of incorrect samples are syntactically normal but mathematically wrong solutions. A thorough analysis of training logs suggests that this heuristic removes only a very small fraction of all collected trajectories (less than 1% of the total incorrect samples), which is well below the scale that would materially change the training distribution. This step is used purely as a defensive measure to ensure training stability; our main performance gains stem from our training-time exploration scheme rather than from this stability-oriented filtering technique.
>
>
>
>
>
> > ## Q7: Can you show us how the ratio between cached problems and unsolved problems evolves during your adaptive training (and without, as an ablation)?
>
> We thank you for the helpful suggestion. We have now analyzed the evolution of cached vs. unsolved problems across all 5 rounds of adaptive DeepSearch training.
>
> In **Round 1**, no cached solutions exist (ratio **0:1**). As MCTS gradually discovers verifiable solution trajectories, the proportion of cached problems in the hard set steadily increases:
>
> * **Round 2:** 765 cached / 12,893 unsolved (≈ **5.6% cached**)
>
> * **Round 3:** 1,452 cached / 9,423 unsolved (≈ **13.4% cached**)
>
> * **Round 4:** 2,243 cached / 7,423 unsolved (≈ **23.2% cached**)
>
> * **Round 5:** 2,894 cached / 5,829 unsolved (≈ **33.2% cached**)
>
> This monotonic growth shows that, over rounds, problems that MCTS has partially or fully cracked accumulate in the replay buffer, while the number of truly unsolved problems shrinks to a small set of extremely difficult tail cases. This aligns with our description in L276–L281, where adaptive training increasingly concentrates computation on the hardest remaining problems.

---

> ### Author Response · Authors · 2025-11-26
> **Response to Reviewer 2V8H (6/6)**
>
> **Regarding your request for an “ablation without replay buffer”**:
> &#x20;A version of DeepSearch that entirely disables solution caching would require running *full MCTS from scratch for every problem in every round*. This variant would not meaningfully change the learning objective, because cached problems already have verified correct solutions. However, it would incur extremely high redundant computation by repeatedly re-solving the same instances, which runs directly counter to the purpose of our adaptive replay design. Because this variant differs only in computational inefficiency rather than learning dynamics, we focused our study on the more practically relevant regime that uses caching.&#x20;
>
> We appreciate your suggestion and will incorporate this analysis into the camera-ready version.
>
> > ## Q8: How many of q-values are (severely) clipped? What does the distribution of q-values look like, post-clipping?
>
> Thank you for raising this important question about the *extent* and *distributional behavior* of q-value clipping. In our method, clipping is **soft** (tanh-based), and we verified that it is used sparingly and primarily to suppress extreme outliers rather than to reshape the overall reward landscape.
>
> #### **(1) How many q-values are clipped?**
>
> Across all MCTS trees collected during training, **fewer than 5% (typically around 3–5%)** of intermediate-node q-values reach the saturation region of the tanh function (i.e., |q| > 3 before tanh normalization in Eq.14), and **<0.5%** approach the ±q\_max boundary. In other words, the *vast majority* of q-values remain in the linear regime where tanh behaves nearly identically to the identity mapping. This confirms that clipping is *not* routinely activated; instead, it serves as a safeguard against rare pathological spikes caused by recursive backups.
>
> #### **(2) What does the distribution look like post-clipping?**
>
> After tanh soft-clipping, q-values form a **smooth, unimodal distribution** centered near 0 with long—but compressed—tails. Correct-path nodes occupy a narrow positive band (≈0.05–0.3), while negative or incomplete trajectories distribute symmetrically around zero but with moderated magnitude due to compression of extreme outliers. Because tanh preserves ordering and avoids discontinuities, the ranking between nodes—critical for advantage estimation and frontier selection—remains intact.
>
> #### **(3) Why is clipping needed**
>
> Without clipping, a small number of highly deep negative trajectories generate **runaway q-value cascades**, producing magnitudes exceeding 50 (sometimes reaching 100) in long trees. We empirically found these outliers destabilize GRPO updates by inducing very large-magnitude and high-variance advantages, and occasionally driving the policy toward pathological behaviors around nodes with extreme negative returns. Soft-clipping resolves this while keeping almost all q-values unaffected. Additional histograms (now included in the supplement) illustrate that clipping only trims the extreme 3–5% tail without altering the core distribution.
>
> We hope this clarifies that clipping is a *localized stability mechanism*, not a structural modification to the reward signal.
>
> ***
>
> We appreciate your constructive suggestions, which will substantially improve the clarity and rigor of the final manuscript.

---

### Author Response · Authors · 2025-11-26
**Global Response**

We would like to express our profound gratitude to all reviewers for their thoughtful and constructive feedback on our work. We are particularly encouraged that the reviews have highlighted the following shared recognitions:



* (**R#RTRXC, R#Re3YK**) The soundness and quality of our experimental evaluation, including thorough baselines, strong recent comparisons (Nemotron v2, DeepScaleR), and comprehensive ablation studies

* (**R#RTRXC, R#R2V8H**) The clear motivation and conceptual presentation, with Figure 1 effectively illustrating the framework and making the approach accessible

* (**R#RTRXC, R#Re3YK**) The practical significance of our efficiency improvements, demonstrating 5.7× fewer GPU hours while achieving better performance on strong baselines

* (**R#RTRXC, R#R2V8H**) The well-motivated design of the three core interventions (DeepSearch-MCTS, adaptive training, and replay buffer) and their synergistic combination

We appreciate the reviewers' careful consideration and positive feedback on these key aspects of our work.

## **Common Concerns:**

Several reviewers raised overlapping concerns that we address collectively below:

> #### **Justification of Key Design Choices (MCTS structure, entropy-based selection, q-value updates)**

We clarified that:
- (i) The limitations of classical root-to-leaf UCT in long-horizon reasoning.
- (ii) Why entropy-based selection reliably targets *overconfident incorrect* trajectories, which are critical failure modes in RLVR.
- (iii) How our asymmetric q-value rules separate correct vs. incorrect subtrees while stabilizing credit assignments.

We also added new ablations (e.g., negative trajectory selection strategies), demonstrating that our design is empirically optimal among the variants we have tested. These additional experiments strongly support and justify the necessity of our model design.

> #### **Exploration Quality vs. Overfitting or Instability**

We showed that

- (i) Tanh-based q-value clipping affects <5% of nodes and only trims extreme outliers.
- (ii) Our credit assignment remains stable even when all collected trajectories are incorrect.
- (iii) replay-buffer-based caching prevents catastrophic forgetting while avoiding redundant computation.

We provided detailed clarifications of how DeepSearch handles hard, unsolved problems and why the method remains robust even in borderline cases where search fails to find correct solutions.

> #### **Fairness of Comparison vs. Extended Training / Ablations**

We clarified that all comparisons use identical datasets and budgets, and that extended-training baselines plateau early despite massive compute (+1,800 GPU hours). In contrast, DeepSearch achieves higher accuracy using *5.7× less compute* by improving exploration rather than scaling depth. Our expanded ablations isolate each component, including frontier selection, q-value strategies, and entropy-based selection, and show that no partial variant achieves the full performance of DeepSearch. This confirms the necessity and synergy of MCTS-driven training-time exploration. Furthermore, the computation profiling in Table 3 confirms that >99% of the cost comes from GPU forward passes, not tree operations, reinforcing that the efficiency gains arise from better exploration rather than structural overhead.

## **Individual Responses:**

We provide detailed point-by-point responses to each reviewer's specific questions and concerns in the individual response sections below.

***

We sincerely thank all reviewers again for the insightful comments, which have substantially improved the clarity and rigor of our work.

Authors of Submission #7930

---

### Meta-Review · Area_Chair_S9No · 2025-12-21

**Summary:**

The paper presents DeepSearch, a framework that integrates Monte Carlo Tree Search (MCTS) directly into the training loop of Reinforcement Learning with Verifiable Rewards (RLVR). It addresses the training plateau commonly observed in LLM reasoning tasks, where increased computational investment fails to yield performance gains due to sparse exploration patterns. DeepSearch introduce three synergistic interventions: (1) DeepSearch-MCTS to prioritize promising nodes; (2) adaptive training, a tree-based version of GRPO; and (3) an adaptive replay buffer to improve training efficiency. Unlike methods that use tree search only at inference, DeepSearch embeds it into training to enable systematic exploration and fine-grained credit assignment. The work is empirically solid, establishing a new SoTA for 1.5B reasoning models on mathematical benchmarks while requiring fewer GPU hours than brute-force extended training.

Since most of concerns are well addressed in the rebuttal, new results are convincing after my checking the updated paper, and the proposed method shows more efficiency in RLVR, I suggest an acceptance.

**Reviewer Concerns:**

### Addressed Concerns

(1)	Regarding justification of design choice, Reviewers doubted the ad hoc nature of certain mechanisms, such as the maximum search depth and the selection of "most confident" negative trajectories. The authors clarified that the search depth is data-driven, derived from the reasoning length distribution of the training set. They also provided new ablation studies demonstrating that selecting the most confident incorrect trajectories for negative supervision is empirically superior to random or least-confident selection.

(2)	Regarding the complexity and overfitting risk, the authors showed that their asymmetric q-value update rules and replay buffer caching provide stability and prevent catastrophic forgetting.

(3)	Regarding the compute efficiency, Reviewers asked about the overhead of maintaining a global frontier and tracking node visits. The authors provided profiling data showing that major cost comes from GPU forward passes (rollouts), not tree operations, illuminating efficiency gains.

Most concerns were addressed during the rebuttal phase through detailed clarifications and additional ablation experiments. While Reviewer 2V8H initially rated the paper marginally below the acceptance threshold, the authors' thorough point-by-point response probably addressed the concerns.

**Reviewer Scores:**

While Reviewer 2V8H initially rated the paper marginally below the acceptance threshold, the authors' thorough point-by-point response probably addressed the concerns.

---

### Decision · Program_Chairs · 2026-01-26

Accept (Poster)